# BEYOND GREEDY: TOWARDS OPTIMAL DEEP CLASSIFICATION TREES

## ABSTRACT

Decision trees are central to interpretable machine learning but face severe scalability challenges. Existing global optimal methods are limited by binary feature selection and shallow tree depths, while traditional heuristic approaches often sacrifice accuracy. To overcome these limitations, this paper introduces a moving-horizon approximate branch-and-reduce method for constructing near-optimal deep classification trees on large-scale datasets with continuous features. This method is based on a bilevel optimization framework, where the upper-level problem is addressed using a branch-and-reduce method, while the lower-level problem is solved recursively. Although the underlying framework guarantees global optimality, we enhance its efficiency for deeper trees by introducing an approximate solution for the lower-level problem, which can be viewed as a lookahead rollout in reinforcement learning. The accuracy is further refined using a low-cost moving-horizon strategy. Extensive experiments demonstrate that the proposed method consistently outperforms existing heuristic baselines in testing accuracy, while maintaining scalability on large datasets compared to global optimal methods.

The decision tree (DT) model is a cornerstone of machine learning, lauded for its interpretable, flowchart-like structure that makes it particularly suitable for classification and regression tasks requiring transparency and comprehensibility (Freitas, 2014; Krzywinski & Altman, 2017). Unlike many black-box models, DTs offer a higher degree of trustworthiness, which is critical in advancing AI for scientific research and high-stakes decision-making (Rudin, 2019; 2022). However, learning an optimal decision tree (ODT) has been classified as an $\mathcal{NP}$-hard problem (Laurent & Rivest, 1976). Traditional algorithms such as `CART` (Breiman et al., 1984), `ID3` (Quinlan, 1986), and `C4.5` (Quinlan, 1993) have been pretty popular due to their simplicity and efficiency, employing greedy heuristics to recursively partition data from the root to the leaves. Although these approaches generate trees efficiently, they fall significantly short of achieving optimality, especially for deeper trees.

Recent advancements in deterministic optimization techniques for learning ODTs have focused on approaches leveraging mixed-integer programming (MIP), satisfiability (SAT) solvers (Hu et al., 2020; Alòs et al., 2023), and dynamic programming (DP) approaches (Nijssen & Fromont, 2007; Van der Linden et al., 2022; van der Linden et al., 2023). Among these, MIP-based methods (Günlük et al., 2021; Zhu et al., 2020; Aghaei et al., 2024) formulate the training process as a mixed-integer optimization problem (Bertsimas & Dunn, 2017) and have achieved notable success by providing global optimization guarantees (optimality gap). However, efficiency remains a major limitation of these methods. For example, while the work of `Quant-BnB` and `RS-OCT` Mazumder et al. (2022); Hua et al. (2022) has demonstrated the ability to generate optimal trees up to depth 3, extending these approaches to deeper trees remains computationally infeasible. It should be noted that although `ConTree` (Brita et al., 2025) is based on DP and MIP, it is built on `Quant-BnB` and, like other MIP-based methods, remains limited to a maximum depth of 4. In contrast, some other scalable DP methods, such as `DL8.5` (Aglin et al., 2020) and `MurTree` (Demirović et al., 2022), leverage advanced caching strategies to enable the learning of relatively deeper trees. Despite these advances, most DP- and SAT-based approaches (Huisman et al., 2024; Lin et al., 2020; Avellaneda, 2020) face inherent drawbacks. They often require binarizing continuous features, which inflates the feature numbers and necessitates approximation techniques from `BinOCT` (Verwer & Zhang, 2019) (some methods directly consider encoding DTs (Shati et al., 2023)). As a result, these methods solve a feature-binarized approximation of the original problem, which causes information loss, and they still struggle to build trees deeper than 5 on large datasets.

Beyond deterministic approaches, several heuristic methods warrant consideration. One notable example is TAO (Carreira-Perpinán & Tavallali, 2018; Carreira-Perpiñán & Zharmagambetov, 2020; Zharmagambetov et al., 2021), which has been reported to yield only modest improvements over CART (Zhu & Shoaran, 2021; Mazumder et al., 2022). More recently, DPDT (Kohler et al., 2025) has been proposed, which frames tree induction as a Markov Decision Process in conjunction with CART. Although DPDT has been shown to outperform CART, its further accuracy gains come at a substantial computational cost, which significantly limits its optimality. Within practical time budgets, the trees it produces remain far less accurate than globally optimal ones. This gap highlights the urgent need for an advanced method that can deliver both high accuracy and scalability for deep decision trees on large-scale datasets.

This paper introduces the Moving-Horizon Approximate Branch-and-Reduce (MHABR) algorithm for deep DTs on large-scale datasets with continuous features, to achieve higher optimality than heuristic methods (particularly CART) while maintaining scalability. Extensive numerical experiments show that MHABR can successfully train trees of depth 8 on datasets with up to 60 million samples, achieving substantially higher optimality than current heuristic methods and better scalability compared to existing global optimal methods. Additionally, we provide an $\alpha$-tuning option to mitigate overfitting, along with an extended variant that further improves optimality, closer to the global optimum.

**Contributions: (1)** We propose a bilevel optimization framework for learning DTs, where the root node parameters are optimized at the upper level, and the subtree parameters are optimized at the lower level. **(2)** Within this framework, we introduce a branch-and-reduce (BR) method for the upper-level problem. When the lower-level problem is solved recursively, the framework guarantees convergence to global optimality. **(3)** To improve efficiency for deep trees, we propose approximating the lower-level problem using CART. Despite the approximation, the algorithm still guarantees global optimality for depth-2 trees. **(4)** We introduce a moving-horizon (MH) technique to iteratively refine branch parameters at low cost, enabling the training of deep trees on large-scale datasets.

**Performance:** The proposed method is evaluated on 57 UCI datasets (Dua & Graff, 2017) with sample sizes ranging from 47 to 60,807,600 and tree depths from 2 to 8, against six baselines.

- **Testing accuracy:** On 51 small datasets (fewer than 10K samples), our method improves testing accuracy by an average of 0.99% over DL8.5. Furthermore, its high efficiency also allows $\alpha$-tuning, which boosts the initial testing results by an additional 2.82%.

- **Scalability:** On 3 medium datasets (10K–1M samples) and 3 large datasets (1M–60M samples), MHABR can successfully handle these, while no existing global optimal methods (even including local optimal methods like LS-OCT) can complete depth-8 tree training within 24 hours for any datasets. Across these datasets, MHABR achieves an average test accuracy 4.94% higher than CART.

- **Optimality:** Our approach guarantees the global optimality for depth-2 trees. At depth 3, it shows an average gap of only **0.58%** (compared to the global optimal solution) across 42 small datasets.

## 1 PRELIMINARY

This paper focuses on training a DT model for a classification task. The notation primarily follows the work of Mazumder et al. (2022). We address a supervised learning problem involving a given dataset $\mathcal{X} = \{x_i \mid x_i \in \mathbb{R}^p, i \in [n]\}$ and corresponding label set $\mathcal{Y}$ that includes $n_c$ classes, where $[n]$ denotes $\{1, \cdots, n\}$ and $n$ is the number of samples, $p$ is the number of features, $x_i$ represents the feature vector that can admit numerical, categorical, or binary values. Initially, we scale each feature value of the dataset to the range $[0, 1]$.

### 1.1 NOTATIONS FOR DECISION TREES

We develop a balanced binary tree formulation with axis-aligned splits and a fixed depth, denoted by $d$. Let $\mathcal{N}_B := \{1, \cdots, 2^d - 1\}$ and $\mathcal{N}_L := \{2^d, \cdots, 2^{d+1} - 1\}$ denote the sets of branch nodes and leaf nodes, respectively. In these trees, each branch node performs a logical test $x_i^a \leq b$, where $x_i^a$ denotes the value of the feature $a$ for the $i$th sample, thereby partitioning the samples into the left (if "yes") and right (if "no") subtrees. The splitting rule at each branch node is defined by two parameters: the feature indices and the corresponding splitting thresholds, collected in the vectors $\mathbf{a}, \mathbf{b}$, respectively. The partitioning operation is repeated at each layer until every sample is assigned

to a leaf node. Each leaf node represents a specific region of the input space and holds a prediction, collected in the vector $\mathbf{c}$.

The induction of a decision tree $T : [0,1]^p \to \mathcal{Y}$ for a group of samples indexed by the set $\mathcal{I}$ can be formulated as an optimization problem expressed as

$$T^*(\mathcal{I}) = \arg\min_{T \in \mathcal{T}_d} L(\mathcal{I}, T), \quad \text{and} \quad V_d(\mathcal{I}) := \min_{T \in \mathcal{T}_d} L(\mathcal{I}, T), \tag{1}$$

where $\mathcal{T}_d$ represents the family of DT with depth $d$ , $L(\mathcal{I}, T) := \sum_{i \in \mathcal{I}} \ell(y_i, T(x_i))$ denotes the number of misclassified instances associated with tree $T$ over the samples $\mathcal{I}$, and $\ell := \mathcal{Y} \times \mathcal{Y} \to \mathbb{R}$ denotes the loss function. For classification tasks in this paper, the loss function is defined as $\ell(y_i, \hat{y}_i) = \mathbb{1}\{y_i \neq \hat{y}_i\}$. In this formulation, a penalty term $\alpha \cdot \mathcal{C}$ can be introduced to balance the trade-off between training accuracy and model generalization, where $\mathcal{C}$ denotes the complexity of the decision tree model and $\alpha$ serves as the regularization coefficient. However, this paper focuses on optimizing the training process; thus, the penalty term is omitted in this formulation. The term will later be incorporated into numerical experiments to assess its impact on model performance.

For a selected feature $a$, let $n_a$ denote the number of unique values in the dataset $x_i^a$, $i \in \mathcal{I}^a$, where $\mathcal{I}^a := [n]$ represents the index set of the data sorted by the values of feature $a$. Define $\mathcal{K}^a := [n_a + 1]$ as the index set of distinct values, denoted as $u_1^a < u_2^a < \cdots < u_{n_a}^a$, with boundary points $u_0^a = u_1^a - 1$ and $u_{n_a+1}^a = u_{n_a}^a + 1$. Since all points within the interval $(u_{k-1}^a, u_k^a)$ are equivalent for splitting, the splitting threshold is defined as $\beta_k^a := \frac{u_{k-1}^a + u_k^a}{2}$ for each $k \in \mathcal{K}^a$. We define $\zeta : \mathcal{K}^a \to \mathcal{I}^a$ such that $\zeta(k)$ is the index of the first occurrence of the unique element $u_k^a$ in the sequence $(x_i^a)_{i \in \mathcal{I}^a}$. For the root node, it suffices to consider candidate trees with all splits located in the set $\mathcal{B} := \{\beta_k^a | a \in [p], k \in \mathcal{K}^a\}$. Given a set $\mathcal{I}^a$ and two integers $k_1$ and $k_2$ such that $0 \leq k_1 \leq k_2 \leq n_a$, we define

$$\mathcal{I}_{[k_1, k_2]}^a := \{i \in \mathcal{I}^a \mid \beta_{k_1}^a \leq x_i^a \leq \beta_{k_2}^a\}, \tag{2}$$

as the set of sample indices $i$ for which the corresponding feature values $x_i^a$ fall within $[\beta_{k_1}^a, \beta_{k_2}^a]$.

## 1.2 A Bilevel Framework for Decision Tree training

We propose a bilevel framework for training decision trees based on the above notations. We denote $T := [a, k, T_L, T_R]$, meaning that $T$ is composed of root node split feature $a$, root node split index $k$ (corresponding to $\beta_k^a$), and the parameters of left and right subtrees $T_L$ and $T_R$. The root node typically exerts the most significant influence on the overall loss in a decision tree (Dwyer & Holte, 2007; Shannon & Banks, 1999), as it processes the entire dataset due to its position at the top of the data flow. Consequently, we isolate the parameters of the root node from the remainder of the tree for optimization through an upper-level problem (*ULP*). The optimal loss is subsequently defined as the sum of the losses from the two subtrees: $T_L$ and $T_R$, optimized in the lower-level problem (*LLP*). Then, the optimization problem for training a depth-$d$ tree $T$ can be denoted as:

$$V_d(\mathcal{I}) := \min_{\substack{a \in [p], \ k \in \mathcal{K}^a \\ T_L, T_R \in \mathcal{T}_{d-1}}} L(\mathcal{I}, [a, k, T_L, T_R]) = \min_{a \in [p], \ k \in \mathcal{K}^a} \left\{ V_{d-1}(\mathcal{I}_{[0,k]}^a) + V_{d-1}(\mathcal{I}_{[k,n_a]}^a) \right\}, \tag{3}$$

as well as an explicit bilevel programming problem:

$$\min_{\substack{a \in [p] \\ k \in \mathcal{K}^a}} \left\{ \min_{T_L \in \mathcal{T}_{d-1}} L(\mathcal{I}_{[0,k]}^a, T_L) + \min_{T_R \in \mathcal{T}_{d-1}} L(\mathcal{I}_{[k,n_a]}^a, T_R) \right\}, \tag{4}$$

where the samples split to the left and right subtrees are denoted by $\mathcal{I}_{[0,k]}^a$ and $\mathcal{I}_{[k,n_a]}^a$. In addition, we use a similar notation to define the optimal value when $a$ is fixed, or when $a$ and $k$ are both fixed.

$$V_d(\mathcal{I}, a) = \min_{k \in \mathcal{K}^a} V_d(\mathcal{I}, a, k) = \min_{k \in \mathcal{K}^a} \left\{ V_{d-1}(\mathcal{I}_{[0,k]}^a) + V_{d-1}(\mathcal{I}_{[k,n_a]}^a) \right\}. \tag{5}$$

# 2 A Branch-and-Reduce Method for Optimal Decision Trees

The bilevel framework, which is explored in a BR method (detailed in Section B), forms the basis of the approach discussed in this section. Here, we present this approach for solving *ULP*, under the key assumption that *LLP* can be solved exactly and efficiently.

Within the *ULP*, the decision variables are $a$ and $k$. The variable $a \in [p]$ can be readily enumerated given the typically limited number of features. However, the selection of $k \in \mathcal{K}^a$ is more complex, contingent upon the number of unique feature values $n_a$. For continuous features, $n_a$ can be as large as $n$, rendering the search space prohibitively extensive for exhaustive enumeration. To mitigate this challenge, we propose a BR method to determine the optimal $k$ in this section.

## 2.1 UPPER BOUND, LOWER BOUND, REDUCTION, AND BRANCHING STRATEGY

The BR method involves four primary steps: Upper Bound, Lower Bound, Reduction Strategy, and Branching Strategy. We now present their details in a general form. Consider a node within the BR process, represented by the set of potential split indices $\mathcal{K}_i = \{k \mid l \le k \le m\}$ $(l, m \in [n_a])$. In the following description, we specially select an index $\bar{k} = \lceil \frac{l+m}{2} \rceil$ (the midpoint of $\mathcal{K}_i$) for branching.

**Upper Bound** The upper bound $U$ denotes the best loss among previously evaluated splits, serving as a historical record throughout the iterative search within the feasible set. It is updated iteratively after evaluating a selected index $\bar{k}$ by:

$$U \leftarrow \min\{U, \ V_d(\mathcal{I}, a, \bar{k})\}, \quad \text{where } V_d(\mathcal{I}, a, \bar{k}) = V_{d-1}(\mathcal{I}^a_{[0,\bar{k}]}) + V_{d-1}(\mathcal{I}^a_{[\bar{k},n_a]}). \tag{6}$$

**Lower Bound** The lower bound is based on a Lipschitz-type condition, which is illustrated by:

**Lemma 2.1.** *For any $\mathcal{I} \subseteq [n]$ and $d \ge 1$, let $k_1$ and $k_2$ be two split indices corresponding to feature $a$. Then we have $|V_d(\mathcal{I}, a, k_1) - V_d(\mathcal{I}, a, k_2)| \le |\zeta(k_1) - \zeta(k_2)|$.*

From Theorem 2.1 (detailed in Section B.3), for a given loss of $\bar{k}$, the lower bound of $\mathcal{K}_i$ is

$$LB := \min\{V_d(\mathcal{I}, a, \bar{k}) - |\zeta(\bar{k}) - \zeta(l)|, V_d(\mathcal{I}, a, \bar{k}) - |\zeta(\bar{k}) - \zeta(m)|\} \tag{7}$$

**Reduction Strategy** Given $V_d(\mathcal{I}, a, \bar{k})$, the reduction strategy aims to reduce the search space by removing $\Delta = \{k \in \mathcal{K}_i \mid V_d(\mathcal{I}, a, k) > U\}$. The following lemma (proved in Section B.4) establishes a reduction strategy in which the candidate set is updated as $\mathcal{K}_i \leftarrow \mathcal{K}_i \setminus \Delta$.

**Lemma 2.2.** *For any $\mathcal{I} \subseteq [n]$ and $d \ge 1$, let $V_d(\mathcal{I}, a, \bar{k})$ be a loss value of Problem (3) corresponding to the split rule $\beta^a_{\bar{k}}$, and an upper bound $U$ (assume $U \le V_d(\mathcal{I}, a, \bar{k})$) of this problem. Define $\delta(\bar{k}) := V_d(\mathcal{I}, a, \bar{k}) - U$. Then, for any $|\zeta(\bar{k}) - \zeta(k)| \le \delta(\bar{k})$, we have $V_d(\mathcal{I}, a, k) \ge U$; so, the optimal split index is not in $\Delta = \{k \mid |\zeta(\bar{k}) - \zeta(k)| \le \delta(\bar{k}), \ k \in \mathcal{K}_i\}$.*

**Branching Strategy** To better expand $\Delta$, the branching strategy bisects the candidate split set $\bar{k}$ at its **midpoint**, ensuring a balanced and effective reduction (Ryoo & Sahinidis, 1996; Tawarmalani & Sahinidis, 2004). We can divide $\mathcal{K}_i$ by the midpoint $\bar{k}$ into two subsets: $\mathcal{K}_L$ and $\mathcal{K}_R$ after reduction.

## 2.2 THE PROCEDURE OF BRANCH-AND-REDUCE METHOD

Now, we are ready to introduce the procedure of our BR method. This method is designed to learn globally optimal depth-$d$ decision trees. The process begins with an initial estimate of the upper bound $U$ and the dataset $\mathcal{I}$. For $a \in [p]$, the algorithm sorts the samples by their feature values to obtain $\mathcal{I}^a$ and constructs the candidate split set $\mathcal{B}^a$. The corresponding index set $\mathcal{K}^a$ contains all split candidates $\beta^a_k$ for $k \in \mathcal{K}^a$, which is then passed to the main loop to obtain the optimal $k$.

The main loop of the BR method solves for $V_d(\mathcal{I}, a)$ with fixed $a$. At each iteration, the steps of this loop are as follows: **(i)** A BR node containing the set $\mathcal{K}_i$ is selected from the entire feasible region of $k$, a step referred to as BR node selection. **(ii)** The midpoint candidate split $\bar{k}$ is then chosen from $\mathcal{K}_i$. The algorithm evaluates $V_d(\mathcal{I}, a, \bar{k})$, and updates the upper bound if the new solution improves upon the current best result. **(iii)** Computing $\delta(\bar{k})$ by the updated upper bound $U$, the reduction strategy narrows down the set $\mathcal{K}_i$. **(iv)** Once the reduced set is determined, branching occurs on the remaining region. While the lower bound from Equation (7) enables the calculation of the optimality gap and termination based on a specified tolerance, to ensure a fair comparison with state-of-the-art methods, the subsequent method employs a complete search strategy with a tolerance of 0. Therefore, we consider terminating the main loop when $\mathbb{K} = \emptyset$. We can easily prove that Algorithm 4 converges to the **global optimum** of Equation (3) based on Theorems 2.1 and 2.2 ( detailed in Section B.5).

## 3 A Moving-Horizon Approximate Method for Deep Trees

In the above bilevel optimization framework, we assume that the calculation of $V_{d-1}$ in *LLP* is solved exactly. Under this assumption, global convergence can be guaranteed for trees of any depth. This condition can be satisfied by recursively calling BR for *LLP*, but it results in an exponential increase in computational time. To address this limitation, this section introduces an approximate method that significantly improves computational efficiency while maintaining high accuracy. Despite the approximation, we show that the global optimality for depth-2 trees is still guaranteed.

### 3.1 An Approximate Problem with *LLP* Approximation

In tree structures, the influence of individual nodes on classification loss generally decreases with depth, as deeper nodes tend to affect fewer data points. This implies that *LLP* has a lesser impact on overall accuracy compared to *ULP*. Consequently, we employ the CART algorithm to obtain an approximate solution for *LLP*. This method offers computational efficiency in addressing the *LLP*, with minor influence on the optimality, but substantially reducing the overall computational demand. Let $W_d(\mathcal{I})$ denote the loss of a depth-$d$ CART tree $\hat{T}$ on dataset $\mathcal{I}$. The CART splitting rule for each branch node typically involves minimizing the combined loss of the potential children (detailed in Section C.1). In our implementation, we use the result of this approximate method to update the upper bound, where the approximate problem is denoted as

$$\hat{V}_d(\mathcal{I}) := \min_{a \in [p], k \in \mathcal{K}^a} \hat{V}_d(\mathcal{I}, a, k) = \min_{a \in [p], k \in \mathcal{K}^a} \{W_{d-1}(\mathcal{I}_{[0,k]}^a) + W_{d-1}(\mathcal{I}_{[k,n_a]}^a)\}. \tag{8}$$

Compared to CART, $\hat{V}_d(\mathcal{I})$ effectively expands the feasible search space of *ULP*, leading to improved accuracy, which is essential for the effectiveness of the following MH procedure. Drawing upon the procedural steps of BR, we can obtain an *approximate branch-and-reduce (ABR)* method to solve $\hat{V}_d(\mathcal{I})$, detailed in Algorithm 2. Then, we have the following lemma (detailed in Section C.3), stating that **ABR must not be worse than CART** in this case.

**Lemma 3.1.** *Suppose there is no reduction strategy in Algorithm 2, let $W_d(\mathcal{I})$ and $\hat{V}_d(\mathcal{I})$ be as Equation (8), then $\hat{V}_d(\mathcal{I}) \leq W_d(\mathcal{I})$ for all $d > 0$.*

**Special case** For depth-1 trees, the greedy method can identify global optimal splits, providing exact solutions for the lower-level subproblems in ABR while solving a depth-2 tree. Consequently, ABR guarantees **global optimality** for depth-2 trees, matching the performance of global optimal methods, as verified by numerical experiments.

### 3.2 A Reinforcement Learning Perspective

ABR can be regarded as an approximate DP method (Bertsekas, 2024), and can be explained by Reinforcement Learning (RL) theories. Following DPDT (Kohler et al., 2025), we model the DT problem as a Markov Decision Process. At each layer, the state is defined by the *set of samples* present at the nodes of that layer. The action at a given layer corresponds to the *set of splitting rules* applied to all nodes in that layer. The reward measures the *loss descent* achieved by the splits, calculated as the difference between the total loss at the previous layer and that at the current layer.

For a depth-$d$ tree, the overall objective is to select splitting rules at all layers to minimize the cumulative loss from the root to the leaves. This can be interpreted as a "cost-to-go" problem:

*1st stage cost + optimal tail problem cost.*

Figure 1 is an illustration of *LLP* approximation. Computing the global optimum requires DP that explores all feasible splits, whose number grows exponentially with tree depth. To make this tractable, *LLP* is approximated in the *value space* (see Section 1.2.3 of Bertsekas (2024)) by constructing a suboptimal policy from two *LLP* subtrees using heuristics such as CART, replacing the *optimal tail problem cost*. This approximation can be further enhanced by starting the heuristic from stage 2, a 2-step lookahead method, which improves performance at the cost of additional computation.

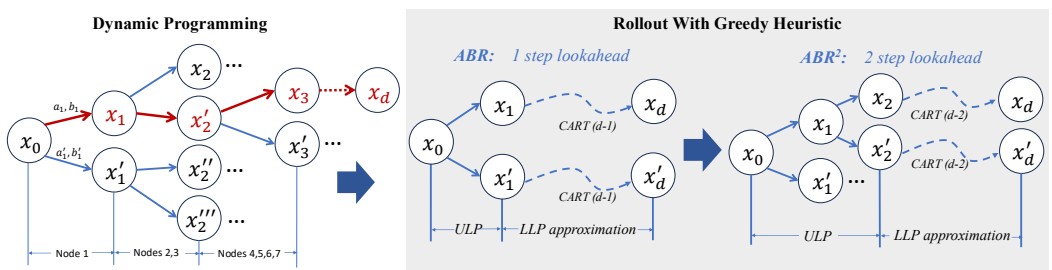

Figure 1: Illustration of the *LLP* Approximation via RL ($x$ denotes the state).

### 3.3 A MOVING-HORIZON APPROACH

The proposed approximation method uses CART to generate suboptimal solutions for the left and right subtrees when $d > 2$. To improve solution quality, we introduce the MH approach that iteratively refines branch parameters. At each node, the subsequent nodes form a subtree that is re-optimized with its corresponding samples $\mathcal{I}_{sub}$, creating a new optimization subproblem. As tree depth increases, this iterative refinement helps close the gap between ABR and the global optimum. As is illustrated in Figure 2, the MH process begins at node 1, where ABR updates all branch nodes in the depth-4 tree ($T_{d=4}$, nodes $1, 2, \ldots, 15$). Next, with node 2 as the root, ABR refines the branch nodes of the depth-3 subtree ($T_{d=3}$, nodes $2, 4, 5, 8, 9, 10, 11$). This iterative procedure continues in each step, with the subtree depth decreasing until the subtree depth $d_{sub} = 2$. Here, nodes such as $4, 8, 9$ in $T_{d=2}$ are updated three times throughout this process.

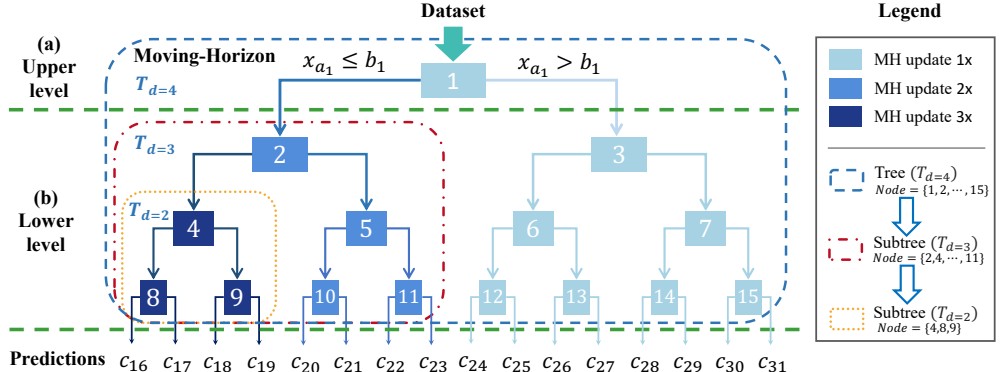

Figure 2: An example of MH when $d = 4$.

After MH slide at any node $t \in \mathcal{N}_B$, the corresponding parameters of the original tree $T$ are updated by $T_{sub}$ as follows

$$T[t \cdot 2^j : (t+1) \cdot 2^j - 1] \Leftarrow T_{sub}[2^j : 2^{j+1} - 1], \quad j \in 0, \ldots, d - d_{sub}. \tag{9}$$

This iterative refinement substantially improves the accuracy of deep decision trees, requiring at most $2^{d-1} - 1$ iterations at relatively low cost, and terminates when a subtree reaches an optimal loss, i.e., $d_{sub} = 1$ or $L(\mathcal{I}_{sub}, T_{sub}) = 0$.

### 3.4 AN MOVING-HORIZON APPROXIMATE BRANCH-AND-REDUCE METHOD

By leveraging the *LLP* approximation and the MH technique, we present the MHABR for a depth-$d$ tree, as described in Algorithm 3. The algorithm iteratively applies the MH procedure for a maximum of $2^{d-1} - 1$ iterations, as it is specifically designed for trees with $d > 2$. For each iteration of MH, a subtree is optimized by ABR (Algorithm 2) to refine the original tree. Now, we give the computational cost of our algorithm (proved in Section C.5).

**Theorem 3.2.** *Suppose there are at most $\tilde{n}$ split candidates at each layer, then the cost of* CART *is bounded by $\mathcal{O}(\tilde{n}d)$, BR is bounded by $\mathcal{O}(\tilde{n}^d)$, and* MHABR *is bounded by $\mathcal{O}(\tilde{n}^2 \frac{d(d-1)}{2})$ for $d \geq 2$.*

**Impact of Reduction in ABR** The reduction strategy may become inaccurate in ABR, as the *LLP* approximation implies that the continuity condition outlined in Theorem 2.1 may not hold. This inaccuracy arises from the approximation $\hat{V}_d(\mathcal{I}, a, k) \approx V_d(\mathcal{I}, a, k)$, which can lead to solutions inferior to those produced by CART, as discussed in Section C.4. Nevertheless, the core principle of MHABR is to expand the *ULP* feasible region, improving accuracy through iterative MH updates. The MH process guarantees improvement by updating the tree only when a re-optimized subtree yields a lower loss. As a result, even when a reduction strategy is used, the resulting optimization problem typically retains a sufficiently large feasible region, allowing for the discovery of lower-loss solutions, as confirmed by numerical experiments. Besides, a warm-start strategy can also be applied before ABR to guarantee better solutions than CART.

---

**Algorithm 1** Main Loop of ABR

1: **function** ABnR$(U, T, \mathcal{I}, d, \mathbb{K})$:
2:   Initialize the iteration index $i = 0$;
3:   **while** $\mathbb{K} \neq \emptyset$ **do**
4:     Node Selection: $\mathcal{K}_i$ from $\mathbb{K}$, set $\mathbb{K} \leftarrow \mathbb{K} \backslash \mathcal{K}_i$, update $i \leftarrow i + 1$;
5:     Select $\bar{k} \in \mathcal{K}_i$, obtain $\mathcal{I}^a_{[0,\bar{k}]}, \mathcal{I}^a_{[\bar{k},n_a]}$;
6:     Calculate $\hat{V}_d(\mathcal{I}, a, \bar{k})$ by Equation (8)
7:     Update Upper Bound:
8:     **if** $\hat{V}_d(\mathcal{I}, a, \bar{k}) < U$ **then**
9:       $U \leftarrow \hat{V}_d(\mathcal{I}, a, \bar{k})$, update $T$;
10:     Reduce: $\delta(\bar{k}) \leftarrow \max\{\hat{V}_d(\mathcal{I}, a, \bar{k}) - U, 1\}$;
11:     Branch: obtain $\mathcal{K}_L, \mathcal{K}_R$ by midpoint;
12:     Update $\mathbb{K} \leftarrow \mathbb{K} \cup \{\mathcal{K}_L\}$, if $\mathcal{K}_L \neq \emptyset$ and update $\mathbb{K} \leftarrow \mathbb{K} \cup \{\mathcal{K}_R\}$, if $\mathcal{K}_R \neq \emptyset$;
13: **return:** Loss $U$ and decision tree $T$.

---

**Algorithm 2** ABR (for subtrees)

1: **function** ABR$(\mathcal{I}, d)$:
2:   Initialize $U$;
3:   **for** $a \in [p]$ **do**
4:     Sort $\mathcal{I}$ to obtain $\mathcal{I}^a, \mathcal{K}^a$, and set $\mathbb{K} \leftarrow \{\mathcal{K}^a\}$;
5:     Calculate $U, T \leftarrow$ ABnR$(U, T, \mathcal{I}, d, \mathbb{K})$;
6: **return:** Loss $U$ and decision tree $T$.

---

**Algorithm 3** MHABR

**input** Dataset $\mathcal{I}$, depth $d$, and initial upper bound $U$;
1: **for** $t \in [2^{d-1} - 1]$ **do**
2:   Get data $\mathcal{I}_t$ at tree node $t$;
3:   **if** $\mathcal{I}_t \neq \emptyset$ **then**
4:     Calculate $U_{sub}, T_{sub} \leftarrow$ ABR$(\mathcal{I}_{sub}, d_{sub})$;
5:     Update $U$ and $T$ through Equation (9);
**output** Loss $U$ and decision tree $T$.

---

## 4 NUMERICAL EXPERIMENTS

This section presents a comprehensive evaluation of our algorithm using benchmark datasets (detailed in Section D). The results compare key performance metrics to 6 baselines, including prediction accuracy and computational efficiency. We provide a comparative analysis with established heuristic methods like CART (Sadeghi et al., 2022), LS-OCT (Dunn, 2018), and DPDT (Kohler et al., 2025), and global optimization methods such as DL8.5 (Aglin et al., 2020) and Quant-BnB (Mazumder et al., 2022), with particular emphasis on their performance with deep trees. Additionally, we use a commercial package IAI-OCT (Interpretable AI, 2025), which is based on (Bertsimas & Dunn, 2017), for comparison. DP-based methods guarantee global optimality on binarized data. Since they differ mainly in efficiency, we report only DL8.5, whose state-of-the-art implementation incorporates advances from MurTree (Demirović et al., 2022).

**Datasets and computing environment:** We collected 57 classification datasets from the UCI Machine Learning Repository (Dua & Graff, 2017), spanning both binary and multi-class tasks, with sample sizes ranging from 47 to 60,807,600. The datasets were categorized into 3 groups: 51 small-scale datasets ($n < 10K$) and 3 medium-scale ($10K \leq n \leq 1M$), and 3 large-scale datasets ($n \geq 1M$). Before the experiments, each dataset was randomly split, with 75% for training and 25% for testing, respectively. The results reflect the average of 10 runs for each dataset. All experiments are conducted on a 40-core *Intel Xeon Gold 5115 CPU* (2.40 GHz) with 93.9 GB RAM.

**Implementation:** Our algorithm is implemented in Julia, with the source code publicly available on GitHub[1]. We also provide an enhanced version of CART (Section C.2), which is developed from DecisionTree.jl package and integrated in MHABR. All the baselines are obtained from their official repository, except LS-OCT, which we reproduced in Julia. Time limits are set to 4 hours for small-scale datasets and 24 hours for larger datasets. More details are given in Section D.2.

---

[1]https://anonymous.4open.science/r/Anonymous_ICLR-CD7A

## 4.1 PERFORMANCE ON SMALL DATASETS

We evaluate our algorithm on 51 small datasets, with results reported in Table 1 and Section A. To further assess optimality performance, we also test on binarized datasets and measure the gap relative to DL8.5. In addition, our algorithm provides an $\alpha$-**tuning** option to mitigate overfitting. This procedure involves selecting an appropriate $\alpha$ value for the penalty term, $\alpha \cdot \mathcal{C}$, which is added to the objective function in Equation (1). To assess the best performance, we compare the tuned MHABR to the tuned CART and IAI-OCT, where IAI-OCT employs a grid search to identify the optimal combination of hyperparameters, yielding fine-tuned results. For both CART and MHABR, $\alpha$ is selected from a set of 21 candidate parameters $\{0, 0.05, \ldots, 1\}$.

Table 1: Average training and testing accuracy on 51 small datasets

| | Depth | Binarized Datasets | | Original Datasets (ODs) | | | | | | Tuned Methods for ODs | | | |
|---|---|---|---|---|---|---|---|---|---|---|---|---|---|
| | | DL8.5 | MHABR$_{bin}$ | Quant-BnB | LS-OCT | CART | DPDT | DPDT(NP) | MHABR | IAI-OCT | CART($\alpha$) | DPDT($\alpha$) | MHABR($\alpha$) |
| Train % | 2 | 83.18* | 83.18 | **84.31*** | 83.90 | 81.84 | 83.54 | 83.78 | **84.31*** | 83.08 | 81.44 | 82.46 | 81.28 |
| | 3 | 87.71* | 86.97 | 89.44* | 87.59 | 85.96 | 88.19 | 88.02 | 88.93 | 86.80 | 85.09 | 86.14 | 87.91 |
| | 4 | 90.79* | 89.89 | / | 90.14 | 89.02 | 90.95 | 91.00 | **92.26** | 89.09 | 87.54 | 88.41 | 90.62 |
| | 8 | 95.89* | 94.79 | / | 96.94 | 96.70 | 97.72 | 98.01 | **98.55** | 93.05 | 93.58 | 93.02 | 95.68 |
| Test % | 2 | 79.18 | 75.14 | 79.89 | 79.67 | 78.63 | 79.52 | 79.64 | 79.91 | 79.82 | 79.27 | 80.46 | **81.28** |
| | 3 | 81.56 | 79.82 | 82.56 | 82.18 | 81.13 | 81.96 | 82.08 | 82.42 | 82.59 | 82.30 | 83.29 | **84.84** |
| | 4 | 82.42 | 81.52 | / | 83.53 | 82.67 | 83.30 | 83.16 | 83.30 | 84.32 | 84.04 | 84.62 | **86.40** |
| | 8 | 82.62 | 83.58 | / | 85.20 | 85.20 | 85.35 | 84.95 | 84.10 | 87.06 | 87.09 | 86.99 | **88.48** |

[1] "*" denotes global optimal solutions.   [2] **Bold** numbers indicate the best result.   [3] $bin.$ denotes on binarized datasets.   [4] Underlined numbers indicate the average over 42 datasets that Quant-BnB completes.   [5] Quant-BnB fails to converge for depths 4 and 8 on most datasets.

**Training accuracy:** As shown in Table 1, MHABR consistently achieves superior training accuracy, outperforming other heuristic methods by **at least 3.15%** on average. Quant-BnB is competitive only at $d = 3$, completing just 42 datasets with an average gap of only **0.58%** above MHABR (see Section A.4) and failing to converge for deeper trees. On binarized datasets, MHABR$_{bin.}$ performs slightly below DL8.5 (with an average optimality gap of 0.68%), yet it is still surpassed by MHABR by 3.85%. Notably, MHABR reaches the **global optimum** at $d = 2$ on both datasets.

**Testing accuracy:** Despite overfitting at $d = 8$, MHABR retains an average advantage of **0.99%** over DL8.5. Furthermore, MHABR is efficient enough to support $\alpha$-**tuning**, which further improves performance by **2.82%**. The $\alpha$-**tuned** MHABR consistently outperforms all baseline methods across all depths, achieving an average improvement of at least 1.8%, and surpasses the $\alpha$-tuned CART by 2.08% in particular.

**Scalability and efficiency:** A key property of MHABR is its scalability, demonstrated by a slower increase in computational cost with tree depth compared to global optimal methods, while maintaining higher accuracy than heuristic baselines. As shown in Figure 3, across the 51 small-scale datasets, Quant-BnB becomes prohibitively expensive for $d \geq 3$, and DL8.5 exhibits a sharp rise in computational cost with increasing depth, particularly between depths 4 and 8, eventually exceeding the runtime of MHABR. In contrast, MHABR exhibits nearly linear growth in computational cost, similar to other heuristic methods, which highlights its superior scalability for deeper trees.

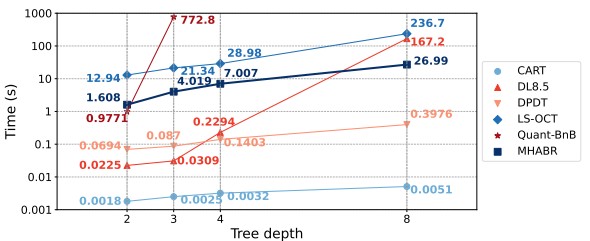

Figure 3: A comparison of training time on 51 datasets.

Table 2: Optimality gap (10 datasets).

| Depth | Method | Accuracy (%) | Gap (%) | Time (s) |
|---|---|---|---|---|
| 3 | CART | 78.99 | 4.29 | 0.01 |
| | MHABR[2] | 82.53* | 0.00 | 0.16 |
| | BR | 82.53* | 0.00 | 0.16 |
| 4 | CART | 81.77 | 4.97 | 0.0005 |
| | MHABR | 85.26 | 0.91 | 0.04 |
| | MHABR[2] | 85.86 | 0.21 | 0.34 |
| | BR | 86.04* | 0.00 | 9.47 |

[1] "*" indicates global optimal solutions.

**Optimality gap evaluation and an extension:** To assess the optimality of our algorithm, we evaluate 10 datasets (from the 51 small datasets) and compute the optimality gap as the relative difference between the solution produced by MHABR and the global optimum obtained by BR. We further introduce an extended variant, MHABR[2] (described in Section 3.2), which improves optimality by extending the *ULP* from the root node to the top two layers (nodes $\{1, 2, 3\}$) for $d \geq 3$. As

summarized in Table 2, MHABR reduces the optimality gap by 3.40% at $d = 3$ and by 4.06% at $d = 4$ relative to CART. Importantly, MHABR[2] attains exact optimal solutions for depth-3 trees and further improves performance for deeper trees.

## 4.2 MEDIUM-SCALE AND LARGE-SCALE DATASETS

This paper addresses the challenge of learning deep trees for large-scale datasets. Table 3 shows results for $d = 4, 8$ using three medium and three large datasets compared to 4 baselines (detailed in Section D.3). For large-scale applications of our algorithm, we incorporate additional techniques to improve the efficiency of our algorithm, as detailed in Section A.6, which also provides an **ablation study** analyzing these techniques, MH, and the reduction strategy.

Table 3: Performance comparison on 3 medium-scale (top) and 3 large-scale datasets (bottom).

| Dataset (n, p, class) | Method | Train (%) d=4 | Test (%) d=4 | Time (s) d=4 | Train (%) d=8 | Test (%) d=8 | Time (s) d=8 | Dataset (n, p, class) | Method | Train (%) d=4 | Test (%) d=4 | Time (s) d=4 | Train (%) d=8 | Test (%) d=8 | Time (s) d=8 |
|---|---|---|---|---|---|---|---|---|---|---|---|---|---|---|---|
| Avila (10430, 10, 12) | CART | 57.24 | 56.58 | 0.03 | 76.77 | 74.87 | 0.05 | SUSY (5000000, 18, 2) | CART | 76.60 | 76.61 | 29.32 | 78.38 | 78.34 | 75.79 |
| | DPDT | 58.71 | 58.85 | 0.90 | 79.71 | 77.00 | 2.96 | | DPDT | 76.86 | 76.87 | 1197.44 | 78.72 | 78.68 | 2951.08 |
| | LS-OCT | 60.33 | 59.87 | 183.61 | 79.84 | 77.00 | 1585.04 | | LS-OCT | / | / | / | / | / | / |
| | DL8.5 | 61.22 | **60.97** | 26.71 | 80.71 | 78.31 | 86400.00 | | DL8.5 | **77.84** | 77.84 | 86400.00 | 76.32 | 76.33 | 86400.00 |
| | MHABR | **61.59** | 60.75 | 8.42 | **93.19** | **91.53** | 14.21 | | MHABR | **77.84** | **77.86** | 17632.26 | **78.94** | **78.89** | 11953.41 |
| Eeg (14980, 14, 2) | CART | 70.12 | 69.30 | 0.03 | 79.31 | 76.30 | 0.06 | HIGGS (11000000, 28, 2) | CART | 65.58 | 65.55 | 81.98 | 69.47 | 69.40 | 216.10 |
| | DPDT | 72.86 | 71.92 | 1.20 | 83.72 | 79.48 | 4.24 | | DPDT | 66.20 | 66.18 | 3223.50 | 69.67 | 69.57 | 7158.52 |
| | LS-OCT | 71.94 | 71.16 | 326.68 | 81.93 | 78.43 | 2498.00 | | LS-OCT | / | / | / | / | / | / |
| | DL8.5 | **74.82** | **73.52** | 40.90 | 79.50 | 75.58 | 86400.00 | | DL8.5 | 66.46 | 66.44 | 86400.00 | 56.75 | 56.77 | 86400.00 |
| | MHABR | 74.50 | 72.84 | 23.98 | **88.93** | **82.51** | 102.00 | | MHABR | **66.88** | **66.85** | 11826.94 | **70.17** | **70.05** | 78370.11 |
| Skin-segmentation (245057, 3, 2) | CART | 97.42 | 97.37 | 0.08 | 99.50 | 99.49 | 0.10 | WESAD (60807600, 8, 8) | CART | 57.46 | 57.46 | 123.23 | 80.27 | 80.26 | 293.24 |
| | DPDT | 98.22 | 98.20 | 5.48 | 99.76 | 99.74 | 10.58 | | DPDT | 61.21 | 61.20 | 5225.47 | 79.45 | 79.45 | 14623.90 |
| | LS-OCT | 98.41 | 98.39 | 1471.57 | 99.75 | 99.73 | 15259.16 | | LS-OCT | / | / | / | / | / | / |
| | DL8.5 | 98.61 | 98.57 | 48.60 | 99.43 | 99.40 | 86400.00 | | DL8.5 | / | / | / | / | / | / |
| | MHABR | **98.72** | **98.72** | 46.62 | **99.95** | **99.91** | 164.83 | | MHABR | **61.33** | **61.33** | 1464.82 | **85.37** | **85.38** | 7977.35 |

[1] **Bold** numbers indicate the best result among all the methods.    [2] LS-OCT fails to converge on large datasets when $d = 8$.
[3] DL8.5 converges only at $d = 4$ on medium datasets; in other cases, it terminates at the time limit or fails to converge.
[4] MHABR is a light version of MHABR incorporating additional techniques.

**Performance on medium and large datasets:** On medium datasets, though DL8.5 achieves accuracy competitive with MHABR at depth-4 trees, it requires more time, and will terminate early when $d = 8$ due to the 24-hour time limit. On large datasets, LS-OCT fails entirely, and DL8.5 also struggles to produce solutions. In contrast, MHABR consistently achieves the best accuracy, with only CART and DPDT able to complete the runs alongside it. MHABR significantly improves testing accuracy over these two heuristic algorithms, outperforming CART by 2.58% at depth 4 and **4.94%** at depth 8, and exceeding DPDT by 0.86% at depths 4 and **4.06%** and at depths 8, respectively. Notably, MHABR delivered exceptional results on dataset Avila for $d = 8$, surpassing other methods by at least 12.4% in both training and testing accuracy.

**Limitation of our method**    While our method demonstrates superior testing performance on small datasets, much of this improvement stems from the $\alpha$-tuning mechanism. Overfitting becomes noticeable at $d = 8$, and redundant computations from repeatedly solving similar subproblems remain an issue. Theoretically, its worst-case performance matches CART, but unpredictable reduction errors (detailed in Section C.4) make formal guarantees, as provided in similar work (Mazumder & Wang, 2023), difficult to achieve. Therefore, our conclusions primarily rely on empirical results.

## 5 CONCLUSIONS AND DISCUSSIONS

In summary, we introduce an approximate BR method for decision tree training within a bilevel optimization framework that separates the tree into upper and lower levels. By leveraging CART to approximate *LLP* and employing the MH procedure to refine branching iteratively, the method achieves superior scalability and consistently higher testing accuracy than other heuristics, aided by $\alpha$-tuning. It guarantees global optimality at $d = 2$, maintains minimal gaps at greater depths, and effectively scales to million-level datasets through mini-batching and an $\varepsilon$-tolerance parameter, with further improvements possible by enlarging the upper level.

ETHICS STATEMENT

This work adheres to the ICLR Code of Ethics. It involves only publicly available datasets, poses no direct risks to individuals or communities, and all experiments are conducted with transparency and research integrity. No conflicts of interest are present.

REPRODUCIBILITY STATEMENT

The authors provide an anonymous, downloadable implementation of the proposed method, allowing full reproduction of all reported results. Detailed information on the computational resources required for each experiment is included, and all datasets used in this work are clearly described with accessible download links.

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

# APPENDIX

# A ADDITIONAL EXPERIMENTS

## A.1 FRIEDMAN TEST AND NEMENYI POST-HOC TEST

We applied the Friedman test to assess whether statistically significant differences exist among the methods across all datasets for each tree depth $2, 3, 4, 8$. This provides a systematic and quantitative framework for method comparison across different depths. To make the comparison more intuitive, we ranked the methods within each dataset (higher accuracy $\Rightarrow$ better rank) and averaged these ranks to compute the mean rank for each method. The mean ranks are presented in Table 4. Based on the $\chi^2$ and $p$-values, $\alpha$-tuned MHABR significantly outperforms the other methods, with the differences becoming more pronounced as tree depth increases.

Table 4: Test accuracy mean ranks on 51 small datasets

| Depth | $\chi^2$ | $p$-value | DL8.5 | Quant-BnB | LS-OCT | CART | DPDT | MHABR | IAI-OCT | CART $(\alpha)$ | MHABR $(\alpha)$ |
|---|---|---|---|---|---|---|---|---|---|---|---|
| 2 | 53.772 | <0.0001 | 5.039 | 5.451 | 4.931 | 6.078 | 5.255 | 5.382 | 5.176 | 5.059 | **2.627** |
| 3 | 95.218 | <0.0001 | 5.098 | 5.549 | 5.373 | 6.373 | 5.873 | 5.510 | 4.804 | 4.539 | **1.882** |
| 4 | 115.537 | <0.0001 | 5.108 | / | 5.147 | 5.882 | 4.961 | 5.569 | 3.912 | 3.716 | **1.706** |
| 8 | 149.571 | <0.0001 | 6.039 | / | 5.343 | 5.235 | 4.980 | 6.108 | 3.480 | 2.873 | **1.941** |

[1] **Bold** numbers indicates the best rank among these methods.

Since the Friedman test indicated significant differences, we performed a Nemenyi post-hoc test for pairwise comparisons, resulting in a p-value matrix. This matrix was visualized as a heatmap, which is shown in Figure 4. The results show that $\alpha$-tuned MHABR consistently outperforms all other methods across tree depths, while tuned methods (IAI-OCT and $\alpha$-tuned CART) gain significantly for $d \geq 4$, and untuned methods show little improvement and begin to overfit.

In conclusion, the scalability of our algorithm, as shown in Figure 3, surpasses that of global optimal methods and allows for $\alpha$-tuning to further enhance testing performance. Combined with its strong optimality, this enables $\alpha$-tuned MHABR to achieve significant superiority over other methods.

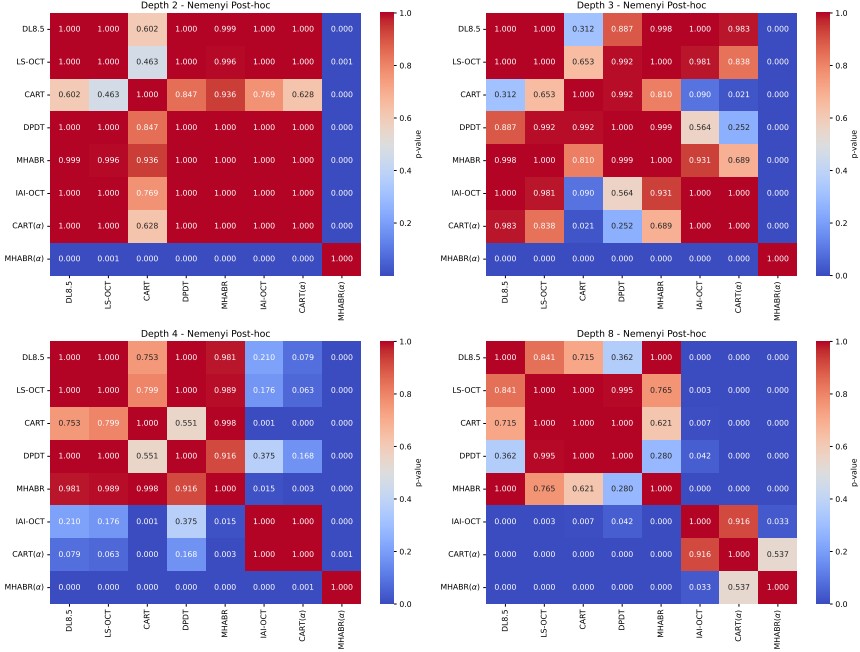

Figure 4: Heat map of Nemenyi Post-hoc test

## A.2 Optimality gap evaluation

We evaluate the optimality gap using 10 small datasets solvable by BR, with results reported in Tables 5 and 6. As shown, CART exhibits a relatively large gap, typically exceeding 3% on average at both depths 3 and 4. In contrast, MHABR maintains a gap within 1% on average, while the extended variant, MHABR$^2$, further improves optimality, consistently reducing the gap below 1%. Correspondingly, the computational cost increases from CART to BR as optimality improves. Therefore, we generally recommend MHABR, which offers strong optimality while keeping the computational cost reasonable.

Table 5: Optimality gap of CART, MHABR, MHABR$^2$, and BR across 10 datasets ($d = 3$)

| Dataset | $n$ | $p$ | $c$ | CART (greedy) | | | MHABR | | | MHABR$^2$ | | | BR (optimal) | | |
|---|---|---|---|---|---|---|---|---|---|---|---|---|---|---|---|
| | | | | Accuracy (%) | Gap (%) | Time ($s$) | Accuracy (%) | Gap (%) | Time ($s$) | Accuracy (%) | Gap (%) | Time ($s$) | Accuracy (%) | Gap (%) | Time ($s$) |
| Iris | 150 | 4 | 3 | 97.14 | 2.68 | 0.0004 | 99.64 | 0.18 | 0.02 | 99.82 | 0.00 | 0.09 | 99.82 | 0.00 | 0.09 |
| Haberman-survival | 306 | 3 | 2 | 79.17 | 3.97 | 0.0002 | 81.88 | 0.69 | 0.01 | 82.45 | 0.00 | 0.12 | 82.45 | 0.00 | 0.12 |
| Monks-problems-3 | 554 | 6 | 2 | 96.48 | 2.41 | 0.0002 | 96.43 | 2.46 | 0.01 | 98.87 | 0.00 | 0.02 | 98.87 | 0.00 | 0.02 |
| Monks-problems-1 | 556 | 6 | 2 | 78.20 | 12.15 | 0.0005 | 88.11 | 1.02 | 0.01 | 89.02 | 0.00 | 0.02 | 89.02 | 0.00 | 0.02 |
| Monks-problems-2 | 600 | 6 | 2 | 66.53 | 2.70 | 0.0019 | 68.07 | 0.45 | 0.01 | 68.38 | 0.00 | 0.02 | 68.38 | 0.00 | 0.02 |
| Balance-scale | 625 | 4 | 3 | 75.94 | 3.45 | 0.0001 | 77.80 | 1.09 | 0.01 | 78.65 | 0.00 | 0.02 | 78.65 | 0.00 | 0.02 |
| Blood-transfusion | 748 | 4 | 2 | 79.73 | 2.36 | 0.0001 | 80.82 | 1.03 | 0.02 | 81.66 | 0.00 | 0.45 | 81.66 | 0.00 | 0.45 |
| Mammographic-mass | 830 | 5 | 2 | 85.27 | 1.16 | 0.0004 | 86.08 | 0.22 | 0.02 | 86.27 | 0.00 | 0.31 | 86.27 | 0.00 | 0.31 |
| Contraceptive-method-choice | 1,473 | 9 | 3 | 52.40 | 9.95 | 0.0010 | 57.61 | 1.00 | 0.03 | 58.19 | 0.00 | 0.51 | 58.19 | 0.00 | 0.51 |
| Car-evaluation | 1,728 | 6 | 4 | 78.99 | 3.72 | 0.0001 | 81.54 | 0.60 | 0.01 | 82.04 | 0.00 | 0.07 | 82.04 | 0.00 | 0.07 |

Table 6: Optimality gap of CART, MHABR, MHABR$^2$, and BR across 10 datasets ($d = 4$)

| Dataset | $n$ | $p$ | $c$ | CART (greedy) | | | MHABR | | | MHABR$^2$ | | | BR (optimal) | | |
|---|---|---|---|---|---|---|---|---|---|---|---|---|---|---|---|
| | | | | Accuracy (%) | Gap (%) | Time ($s$) | Accuracy (%) | Gap (%) | Time ($s$) | Accuracy (%) | Gap (%) | Time ($s$) | Accuracy (%) | Gap (%) | Time ($s$) |
| Iris | 150 | 4 | 3 | 99.46 | 0.54 | 0.0005 | 100.00 | 0.00 | 0.02 | 100.00 | 0.00 | 0.30 | 100.00 | 0.00 | 8.36 |
| Haberman-survival | 306 | 3 | 2 | 80.79 | 6.94 | 0.0002 | 85.33 | 1.71 | 0.03 | 86.64 | 0.20 | 0.24 | 86.81 | 0.00 | 5.05 |
| Monks-problems-3 | 554 | 6 | 2 | 98.87 | 0.00 | 0.0002 | 98.87 | 0.00 | 0.02 | 98.87 | 0.00 | 0.04 | 98.87 | 0.00 | 0.24 |
| Monks-problems-1 | 556 | 6 | 2 | 81.77 | 16.26 | 0.0006 | 95.23 | 2.48 | 0.02 | 97.58 | 0.07 | 0.04 | 97.65 | 0.00 | 0.25 |
| Monks-problems-2 | 600 | 6 | 2 | 69.42 | 4.46 | 0.0021 | 71.84 | 1.13 | 0.03 | 72.20 | 0.64 | 0.05 | 72.67 | 0.00 | 0.27 |
| Balance-scale | 625 | 4 | 3 | 80.11 | 5.66 | 0.0001 | 84.25 | 0.78 | 0.03 | 84.57 | 0.40 | 0.06 | 84.91 | 0.00 | 0.39 |
| Blood-transfusion | 748 | 4 | 2 | 80.41 | 4.00 | 0.0001 | 82.73 | 1.23 | 0.08 | 83.35 | 0.49 | 1.08 | 83.76 | 0.00 | 34.71 |
| Mammographic-mass | 830 | 5 | 2 | 85.71 | 2.36 | 0.0005 | 87.23 | 0.62 | 0.07 | 87.59 | 0.22 | 0.73 | 87.78 | 0.00 | 15.38 |
| Contraceptive-method-choice | 1,473 | 9 | 3 | 57.32 | 5.76 | 0.0011 | 60.10 | 1.19 | 0.11 | 60.66 | 0.27 | 1.40 | 60.82 | 0.00 | 28.79 |
| Car-evaluation | 1,728 | 6 | 4 | 83.80 | 3.82 | 0.0001 | 87.04 | 0.11 | 0.05 | 87.13 | 0.00 | 0.15 | 87.13 | 0.00 | 1.24 |

## A.3 Compared to DL8.5

In the numerical experiments, we present the results of three methods across 51 small datasets. DL8.5 is a global optimal method, however, on binarized datasets. The error impact can be found in Table 1. In the following, we provide a more detailed analysis of its outcomes, using box plots to illustrate the statistical metrics for these results, as shown in Figure 5.

Regarding the training outcomes, MHABR shows comparable performance to DL8.5 for $d \in \{2, 3, 4\}$, exhibiting marginally superior results in some instances. Notably, when $d = 8$, MHABR significantly outperforms the other two methods, achieving a high degree of accuracy approaching 100%.

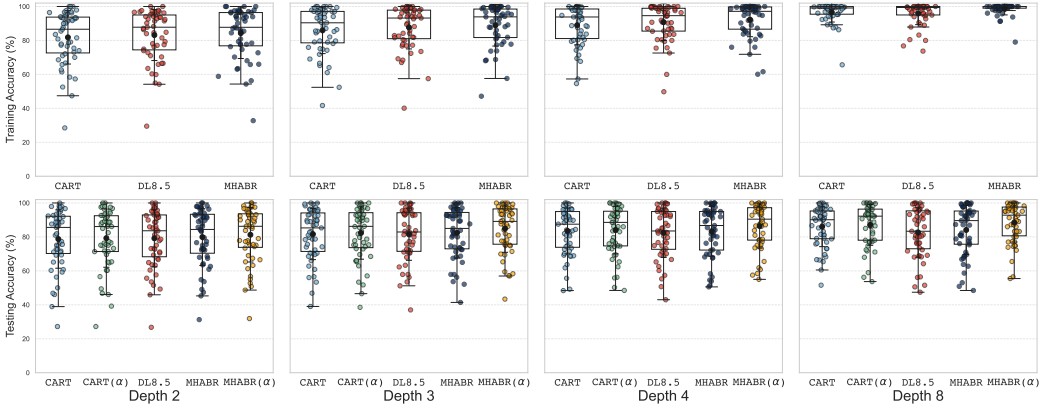

Figure 5: Training and testing accuracy of 51 small datasets across $d = 2, 3, 4, 8$.

The testing results reveal that the $\alpha$-tuned MHABR consistently outperforms the other evaluated methods. This is evidenced by its corresponding boxes consistently exhibiting the highest proximity

to 100% accuracy across all considered cases among the five methods. This observation suggests that the $\alpha$-tuning procedure effectively enhances the testing accuracy of MHABR.

## A.4 COMPARED WITH QUANT-BNB

This subsection presents a comparative analysis of MHABR and Quant-BnB, focusing on a subset of 42 small datasets for which all evaluated methods successfully identified feasible solutions. Prior to discussing the comparative performance against Quant-BnB, we delineate the primary distinctions between the two approaches.

Table 7: Comparison on 42 small datasets (excluding 9 datasets Quant-BnB that fail to complete).

|  | Depth | CART | DL8.5 | Quant-BnB | MHABR |
|---|---|---|---|---|---|
| Train (%) | $d = 2$ | $84.10 \pm 14.11$ | $83.93 \pm 15.04^*$ | $\mathbf{84.70 \pm 14.76}^*$ | $\mathbf{84.70 \pm 14.76}^*$ |
|  | $d = 3$ | $87.99 \pm 12.41$ | $88.08 \pm 13.18^*$ | $\mathbf{89.44 \pm 12.36}^*$ | $88.86 \pm 12.62$ |
| Test (%) | $d = 2$ | $78.98 \pm 16.37$ | $79.89 \pm 16.40$ | $\mathbf{80.53 \pm 15.71}$ | $80.47 \pm 15.91$ |
|  | $d = 3$ | $81.41 \pm 14.97$ | $81.71 \pm 15.79$ | $\mathbf{82.56 \pm 15.22}$ | $82.30 \pm 15.14$ |

[1] **Bold** numbers indicate the best solutions.    [2] "*" signifies the global optimum in the corresponding formulation.
[3] Even though Quant-BnB and MHABR can obtain the optimal loss when $d = 2$, the optimal trees might differ.

**Difference from Quant-BnB** As explained in Section 2, our BR method is similar to Quant-BnB, including the same upper bound that is a very common one, and a similar reduction strategy that we generalize to their lower bound 1 for any-depth trees. On the other hand, MHABR is more different when $d \geq 3$ as we introduce the *LLP* approximation, which improves efficiency but simultaneously sacrifices the optimality. However, this sacrifice is worthwhile because we break the depth limitation of Quant-BnB (max $d = 3$) and DL8.5 (max $d = 4$), and our method can provide sufficiently better solutions than CART when $d = 8$ (higher depth would lose interpretability).

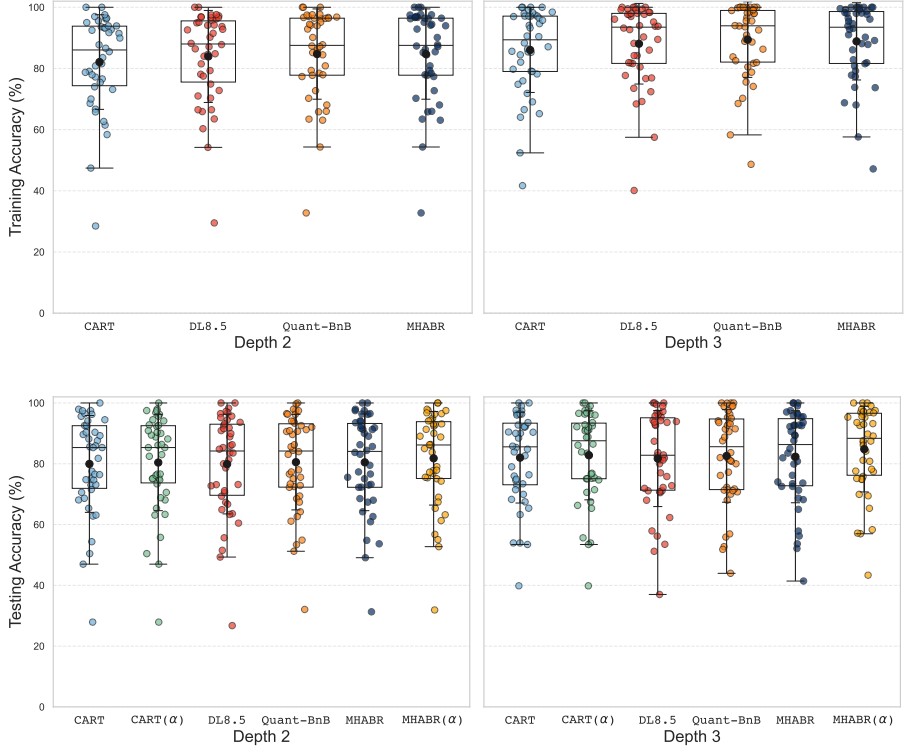

Figure 6: Training and testing accuracy of 42 small datasets across $d = 2, 3$.

The results are presented in Figure 6. `MHABR` achieves equivalent training accuracy to `Quant-BnB` at a depth of $d = 2$, and demonstrates a minor disparity, performing marginally lower than `Quant-BnB`, at $d = 3$. This outcome suggests that the solution generated by `MHABR` closely approximates the optimal solution for shallower trees. Furthermore, it somehow indicates that the potential loss of optimality resulting from the *LLP* approximation is effectively compensated for by the MH procedure.

## A.5 COMPARED TO `DPDT`

`DPDT` serves as a strong heuristic baseline in our comparison, with its scalability largely attributed to parallel acceleration. To assess its optimality, we focus on the case of $d = 8$. The training outcomes are summarized in Table 8. Across 51 datasets, `MHABR` strictly outperforms `DPDT` on 23 datasets with an average improvement of 46.29%, while being only 3.93% worse on 7 datasets. These results demonstrate that `MHABR` achieves substantially stronger optimality compared to `DPDT`.

Table 8: Comparison between `MHABR` and `DPDT` at $d = 8$ across 51 small datasets

| Category ($d = 8$) | Count | Outperform (sum of acc/%) |
|---|---|---|
| MHABR > DPDT | 23 | 46.29 |
| MHABR < DPDT | 7 | -3.93 |
| MHABR = DPDT | 21 | 0.00 |

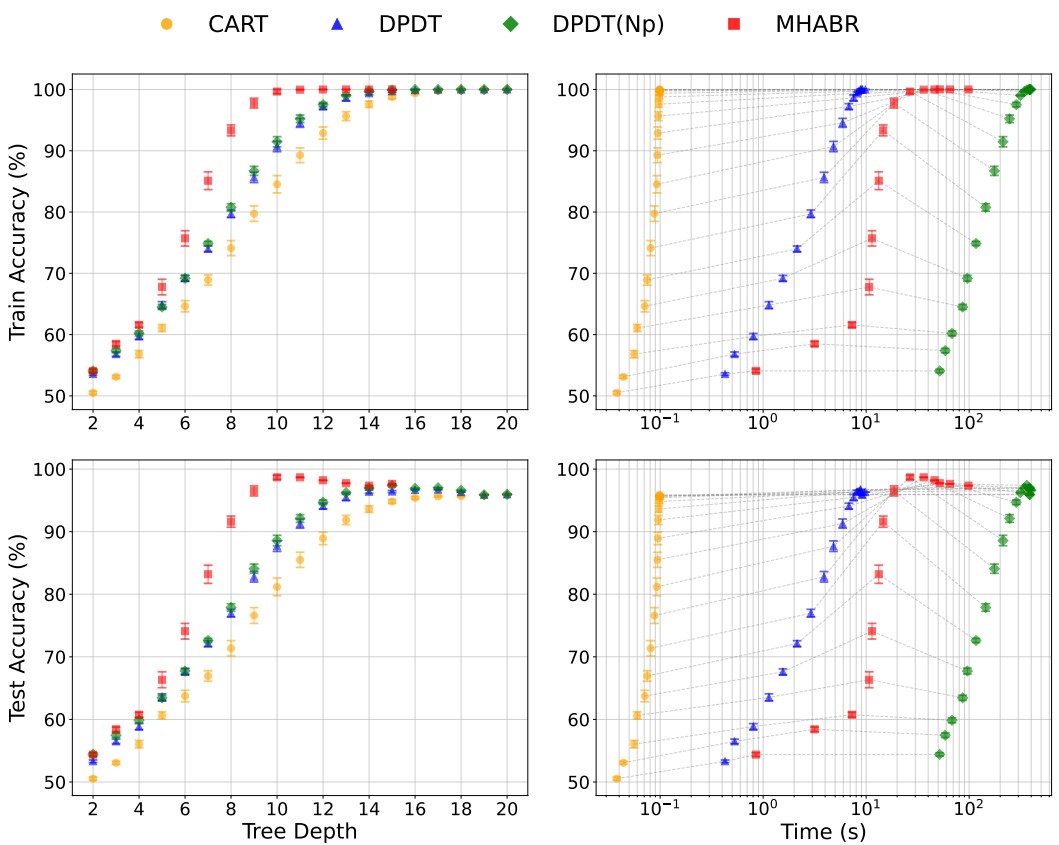

Figure 7: Training/testing accuracy vs. tree depth/running time on `Avila`

For a detailed comparison with `DPDT` under comparable settings, we introduce a new DPDT configuration (denoted by `DPDT(NP)`), `cart_nodes_list=(Np, )`, which enables it to achieve the same theoretical accuracy as `ABR`. To illustrate the performance gains brought by the reduction and MH techniques, we use the datasets `Avila` and `Eeg` to plot the corresponding Pareto fronts

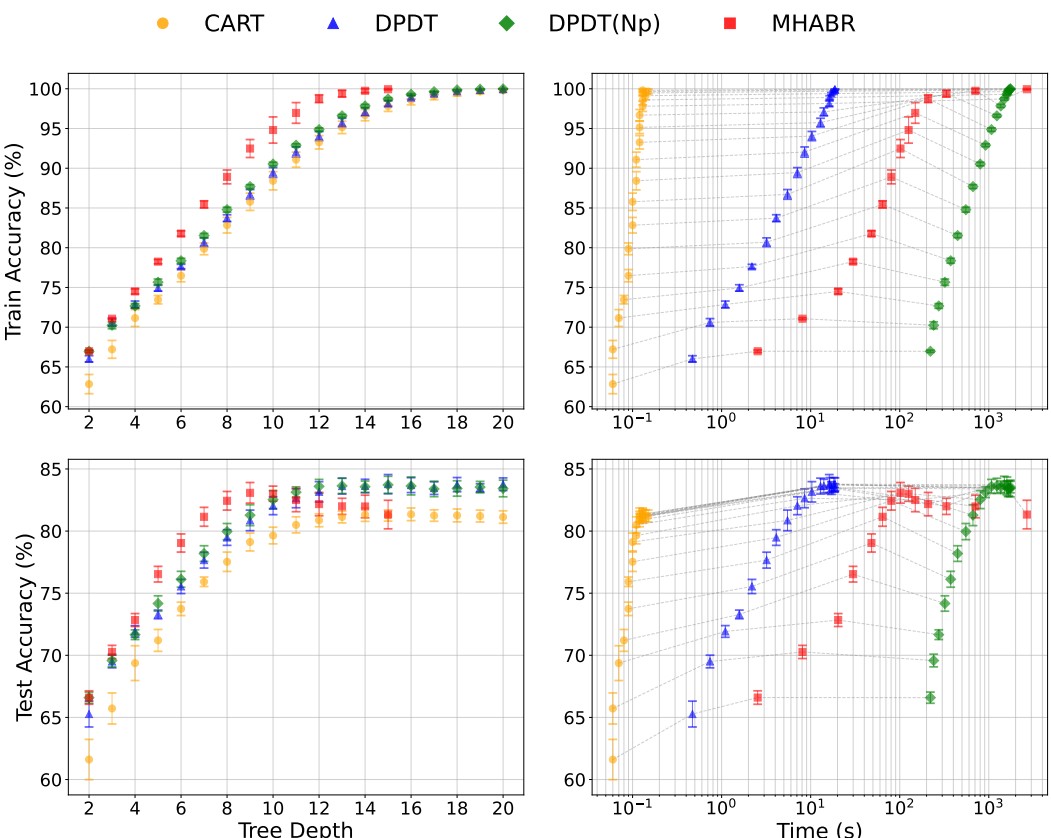

Figure 8: Training/testing accuracy vs. tree depth/running time on `Eeg`

for `CART`, `DPDT`, `DPDT(NP)`, and `MHABR`. We evaluate tree depths from 2 to 20. For `MHABR`, we report results only up to $d = 15$, since it already reaches almost $100\%$ training accuracy at depth 10 and begins to show signs of overfitting thereafter.

The Pareto fronts shown in Figures 7 and 8 illustrate the trade-offs between training/testing accuracy and tree depth, as well as between training/testing accuracy and training time. To clearly show the relationship between tree depth and running time, we also connect the points of each same depth in the figures of training/testing accuracy vs. time. In Figure 7, among all methods, `MHABR` exhibits the strongest overall accuracy. It achieves almost perfect training accuracy $99.66\%$ at depth $d = 10$, while its testing accuracy also achieves the highest peak performance $98.68\%$. Beyond this regime, its testing accuracy declines, reflecting its tendency to overfit at larger depths. In contrast, `DPDT` and `DPDT(Np)` approach this level at least $d \geq 14$, `CART` at least $d \geq 17$. In Figure 8, `MHABR` shows weaker testing accuracy when $d \geq 10$, which is expected due to overfitting, as its training accuracy is nearly 100%. This leads to a drop in testing accuracy as well as the running time increases. Nonetheless, before overfitting occurs, `MHABR` still clearly outperforms the other methods.

In terms of running time, `CART` is the fastest method, but it requires substantially deeper trees (i.e., much larger models) to achieve competitive accuracy. `DPDT` improves accuracy relative to `CART` with only a modest increase in computation, but its strength is mainly evident in its default configuration. For achieving higher accuracy, `MHABR` performs better and requires less time than `DPDT(Np)`. Since model size is crucial for the interpretability of decision trees, obtaining high accuracy with a smaller model is particularly desirable, which is an advantage offered by `MHABR`.

`MHABR` achieves higher accuracy than `DPDT(Np)` at the same depth (before overfitting), demonstrating the substantial performance gains contributed by the MH component. Moreover, `MHABR` requires less training time than `DPDT(Np)` before overfitting occurs, highlighting its superior efficiency enabled by the Reduction technique. Although the slope of the running-time increase for `MHABR`

becomes steeper than that of `DPDT(Np)` once the depth exceeds 13, an effect attributable to the MH procedure, consistent with our complexity analysis, the training accuracy has already reached 100% at this point and cannot improve further. Thus, MH iterations can be relaxed or omitted beyond this depth.

Furthermore, to better assess performance in a realistic setting, we split the data into 50% for training, 25% for validation, and 25% for testing to evaluate the effect of $\alpha$-tuning (with $\alpha \in [0.0, 0.001, 0.005, 0.01, 0.05, 0.1, 0.2]$). The results for 51 small datasets across depths 2–5 are summarized in Table 9. As shown, `MHABR` consistently outperforms the other methods in training and testing accuracy at all depths (more obvious in training) when $d \leq 4$, but achieves slightly lower test accuracy than `DPDT` and `DPDT(Np)` when $d = 5$. This is likely due to the extremely small size of some datasets, where training accuracy reaches 100% and overfitting becomes substantial. To provide a fairer evaluation, we additionally report results on 25 datasets with more than 500 samples in Table 10, where `MHABR` shows improved testing performance at $d = 5$, outperforming the other methods.

Table 9: $\alpha$-tuning results of 51 small datasets.

| Depth | Training Accuracy | | | | Testing Accuracy | | | |
|---|---|---|---|---|---|---|---|---|
| | CART | DPDT | DPDT(Np) | MHABR | CART | DPDT | DPDT(Np) | MHABR |
| 2 | 81.04(2.19) | 83.15(2.11) | 83.24(2.04) | **84.34**(1.50) | 77.91(3.43) | 79.05(3.39) | 79.10(3.35) | **79.66**(3.39) |
| 3 | 84.58(2.61) | 87.14(2.36) | 87.14(2.22) | **88.74**(1.55) | 80.85(3.08) | 81.79(2.89) | 81.71(2.96) | **82.13**(3.27) |
| 4 | 86.88(2.74) | 89.24(2.88) | 89.42(2.76) | **91.69**(1.67) | 82.37(2.95) | 82.68(3.06) | 82.64(3.06) | **83.06**(3.55) |
| 5 | 88.19(3.00) | 91.05(3.30) | 91.29(3.05) | **93.94**(1.53) | 83.27(3.04) | 83.72(3.22) | **83.91**(3.23) | 83.40(3.56) |

Table 10: $\alpha$-tuning results of 25 small datasets ($n \geq 500$).

| Depth | Training Accuracy | | | | Testing Accuracy | | | |
|---|---|---|---|---|---|---|---|---|
| | CART | DPDT | DPDT(Np) | MHABR | CART | DPDT | DPDT(Np) | MHABR |
| **2** | 81.29 (1.19) | 82.54 (1.13) | 82.57 (1.06) | **82.89** (1.03) | 80.07 (1.94) | 81.05 (1.87) | 81.09 (1.93) | **81.34** (3.16) |
| **3** | 84.29 (1.49) | 85.69 (1.33) | 85.78 (1.19) | **86.54** (1.12) | 82.57 (1.85) | 83.47 (1.61) | 83.54 (1.71) | **83.92** (1.77) |
| **4** | 86.66 (1.44) | 87.93 (1.56) | 88.23 (1.23) | **89.20** (1.53) | 84.43 (1.79) | 84.87 (1.69) | 84.77 (1.76) | **85.40** (1.90) |
| **5** | 88.35 (1.91) | 90.27 (2.12) | 90.47 (1.90) | **91.76** (1.77) | 85.42 (2.02) | 86.11 (2.05) | 86.26 (2.12) | **86.31** (1.95) |

## A.6 ABLATION STUDIES

### A.6.1 TECHNIQUES FOR LARGE-SCALE DATASETS: MINI-BATCH AND TOLERANCE TERMINATION

The mini-batch technique uses a fraction $\theta$ of the dataset partitioned into the subtrees for training, with a tolerance $\varepsilon$ as the stopping criterion. While this technique may slightly reduce accuracy, it significantly reduces computational time. The specific steps and analysis for implementation are detailed as follows.

To enhance efficiency for large-scale datasets, mini-batch sampling and an $\varepsilon$-tolerance termination criterion are employed. Mini-batch sampling uses a subset of the dataset, defined by a sampling ratio $\theta \in (0, 1]$, as input to ABR, while the $\varepsilon$-tolerance criterion determines termination. Although this approach may slightly reduce accuracy, it significantly decreases computational time, as demonstrated later. The primary factor affecting computational time is the number of splits in the initial branch, influenced by dataset characteristics, with continuous features often contributing more distinct values, many of which minimally impact the loss function.

For the mini-batch strategy, a proportion $\theta$ of the original data is sampled for each subtree ($T_L$ and $T_R$ in Equation (8), corresponding to the data partitioned to nodes 2 and 3). On large-scale datasets, continuous features often require fewer samples, as their splits generally result in only small fluctuations in the overall loss.

To further enhance the efficiency of this algorithm for extremely large-scale datasets, we introduce a tolerance parameter $\varepsilon$ as a termination criterion. Let $\texttt{length}(\mathbb{K})$ denote the size of the split index set. The termination criterion can be formally defined as:

$$\texttt{length}(\mathbb{K})/n \le \varepsilon. \tag{10}$$

For a fixed feature index $a$, the candidate set of the BR method, denoted by $\mathcal{K}^a$, contains $n_a$ elements. If no element is reduced during the process, the bisection branching strategy generates a set $\mathbb{K}$ with at most $n_a/2^j$ elements after the $j$-th iteration. By Equation (10), the algorithm will terminate after at most $\lceil \log \frac{n_a}{n\varepsilon} \rceil$ iterations. Since the algorithm generates two subsets per iteration, the total number of splits evaluated throughout this process is $\texttt{length}(\mathbb{K}) \le \varepsilon \cdot n$ when the termination criterion is satisfied.

### A.6.2 ABLATION STUDY ON ADDITIONAL TECHNIQUES, MH, AND REDUCTION STRATEGY

We now study how the reduction strategy, the MH method, and tunable parameters $\varepsilon$, $\theta$ affect the computational efficiency and accuracy of Algorithm 3, by systematically evaluating MH procedure performance per layer using an `Avila` case study on a depth-8 decision tree.

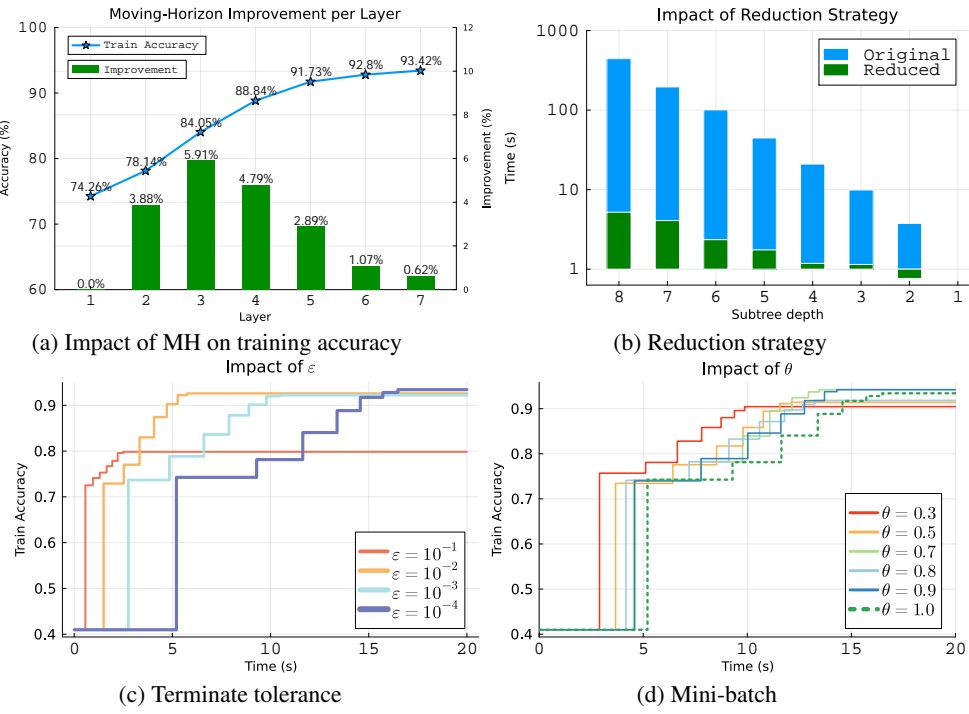

(a) Impact of MH on training accuracy
(b) Reduction strategy
(c) Terminate tolerance
(d) Mini-batch

Figure 9: Impact of the MH procedure, reduction strategy, and parameters $\varepsilon$ and $\theta$. We train a depth-8 tree for `Avila`, where (c) and (d) systematically evaluate MH performance for different subtree depths. For instance, the depth-8 subtree represents the complete original tree, while the depth-7 subtrees are rooted at nodes 2 and 3. Similarly, the depth-6 subtrees are rooted at nodes 4, 5, 6, and 7, and this pattern continues for the other depths.

To evaluate the contribution of the MH refinement, we show the results of a depth-8 tree on the Avail dataset with, without any other techniques except the approximation. The result is shown in Figure 9(a). We observe that the most significant improvement occurs within the first five layers. This suggests that the MH refinements have a greater impact on earlier branch nodes, where the corresponding subtrees are relatively larger and contain more samples. It demonstrates that the reduction strategy significantly accelerates the computation of subtrees at various depths within the MH procedure. It is evident that the reduction strategy yields a decrease in runtime of nearly two orders of magnitude for the depth-8 subtree. While this reduction in computational time becomes less pronounced as the subtree depth decreases, it still contributes to a considerable overall cost reduction.

Figure 9(b) demonstrates that the reduction strategy significantly accelerates the computation of subtrees at various depths within the MH procedure. It is evident that the reduction strategy yields a decrease in runtime of nearly two orders of magnitude for the depth-8 subtree. While this reduction in computational time becomes less pronounced as the subtree depth decreases, it still contributes to a considerable overall cost reduction.

Figure 9(c) illustrates the convergence of results obtained through iterative refinement using the MH method under varying termination conditions. The final accuracies achieved with $\varepsilon = \{10^{-2}, 10^{-3}, 10^{-4}\}$ are notably similar, but $\varepsilon = 10^{-2}$ results in 60% reduction in running time. The scenario with $\varepsilon = 10^{-1}$ exhibits a discernible decrease in accuracy. Overall, larger values of $\varepsilon$ lead to lower accuracy for MHABR, but with the benefit of decreased training time. This suggests the possibility of identifying a suitable $\varepsilon$ that incurs a minor reduction in accuracy while yielding a significant saving in computational time, as exemplified by $\varepsilon = 10^{-2}$ in this case.

The influence of batch sizes is explored in Figure 9(d). Utilizing mini-batches involves training a model on a subset of the data, which invariably reduces computational time. Interestingly, this approach can sometimes yield superior performance. As depicted in the figure, the accuracies achieved with batch size ratios ($\theta$) of 0.8 and 0.9 surpass those obtained when using the complete dataset. Conversely, a batch size ratio of $\theta = 0.3$ results in a relatively discernible reduction in accuracy (but less than 5%).

## B  DETAILS ABOUT PROPOSED ALGORITHMS

This section details the BR method and proves the lemmas and theorems in Section 2.

### B.1  THEORETICAL OPTIMALITY RELATIONSHIPS

First, we illustrate the scope of our method in Figure 10, which summarizes the theoretical relationships among the proposed approaches. As noted in Theorem 3.1, our algorithm guarantees solutions no worse than CART when no reductions are applied, with the MH procedure further refining results toward better optimality. Expanding the ULP scope improves the approximation, making $ABR^2$ superior to ABR, and further enlargement brings methods closer to global optimal approaches such as the unapproximated BR and Quant-BnB. For binarized datasets, the binarization process introduces errors that reduce optimality, so $MHABR_{bin.}$ performs worse than the original dataset version and is clearly inferior to global optimal methods such as DL8.5 and MurTree.

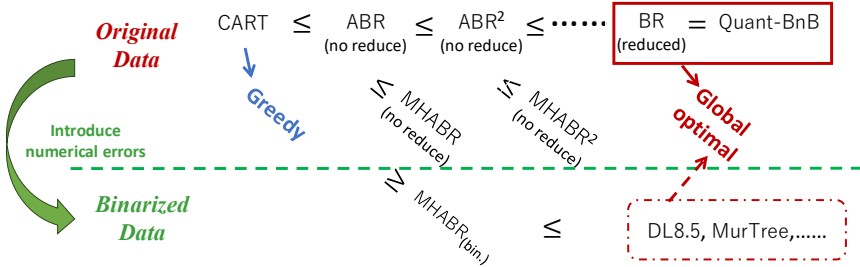

Figure 10: Theoretical relationships between our algorithms and other global optimal methods

### B.2  BRANCH-AND-REDUCED ALGORITHM BASED ON THE BILEVEL FRAMEWORK

The BR method is explored in Algorithm 4 (the orange part for lower bound and optimality is optional), including two functions: $BR(\mathcal{I}, d)$ and $BnR(U, T, \mathcal{I}, d, \mathbb{K})$. In this procedure, $BR(\mathcal{I}, d)$ enumerates $a \in [p]$ and sorts the samples by the corresponding feature value to obtain the initial candidate set $\mathbb{K}$. This set is delivered to the main loop function $BnR(U, T, \mathcal{I}, d, \mathbb{K})$.

Then we execute the branch and reduce procedure, which can be described as:

1. Find the midpoint and evaluate it to obtain the upper bound.

---

**Algorithm 4** Branch-and-reduce method (BR)

---

1: **function** BR($\mathcal{I}, d$):
2:    Initialize $U$;
3:    **for** $a \in [p]$ **do**
4:       Sort $\mathcal{I}$ to obtain $\mathcal{I}^a$, $\mathcal{K}^a$, and set $\mathbb{K} \leftarrow \{\mathcal{K}^a\}$;
5:       $U, T \leftarrow$ BnR($U, T, \mathcal{I}, d, \mathbb{K}$);          *# solve for optimal trees*
6: **return:** Optimal loss $U$ and decision tree $T$.

---

7: **function** BnR($U, T, \mathcal{I}, d, \mathbb{K}$):
8:    Initialize the iteration index $i = 0$;
9:    **while** $\mathbb{K} \neq \emptyset$ **do**
10:      **node selection**
11:        Select a set $\mathcal{K}_i$ from $\mathbb{K}$, set $\mathbb{K} \leftarrow \mathbb{K} \backslash \mathcal{K}_i$;     *# NODE SELECTION*
12:        update $i \leftarrow i + 1$;
13:      **upper bound**
14:        Select a point $\bar{k} \in \mathcal{K}_i$, obtain $\mathcal{I}^a_{[0,\bar{k}]}, \mathcal{I}^a_{[\bar{k},n_a]}$;
15:        $(V_{d-1}(\mathcal{I}^a_{[0,\bar{k}]}),\ T_L) \leftarrow$ BR($\mathcal{I}^a_{[0,\bar{k}]},\ d-1$);     *# recursion for subtrees*
16:        $(V_{d-1}(\mathcal{I}^a_{[\bar{k},n_a]}),T_R) \leftarrow$ BR($\mathcal{I}^a_{[\bar{k},n_a]}, d-1$);
17:        $V_d(\mathcal{I}, a, \bar{k}) = V_{d-1}(\mathcal{I}^a_{[0,\bar{k}]}) + V_{d-1}(\mathcal{I}^a_{[\bar{k},n_a]})$;     *# Evaluation*
18:        **if** $V_d(\mathcal{I}, a, \bar{k}) < U$ **then**
19:           $U \leftarrow V_d(\mathcal{I}, a, \bar{k})$, update $T$;     *# update tree and upper bound*
20:      **lower bound**
21:        Calculate $LB$ by Equation (7);     *# LOWER BOUND*
22:        Obtain the optimality gap $U - LB$;
23:      **if** $U - LB \leq$ tol **then**
24:           **break**     *# given* tol
25:      **reduction**
26:        $\delta(\bar{k}) \leftarrow V_d(\mathcal{I}, a, \bar{k}) - U$;     *# REDUCTION*
27:      **branching**
28:        Obtain $\mathcal{K}_L, \mathcal{K}_R$ by $\bar{k}$;
29:        Update $\mathbb{K} \leftarrow \mathbb{K} \cup \{\mathcal{K}_L\}$, if $\mathcal{K}_L \neq \emptyset$;     *# BRANCHING*
30:        Update $\mathbb{K} \leftarrow \mathbb{K} \cup \{\mathcal{K}_R\}$, if $\mathcal{K}_R \neq \emptyset$;
31: **return:** Optimal loss $U$ and decision tree $T$.

---

  2. Calculate the lower bound and the optimality gap, check the termination condition.
  3. Reduce a circular range using Theorem 2.2 to obtain two subsets.
  4. Recursively apply the process to these subsets.

Without the optional steps, the BnR iterative process terminates upon the evaluation of all candidate split points, i.e., when $\mathbb{K} = \emptyset$. Conversely, when incorporating the optional steps, Algorithm 4 functions as a standard branch-and-bound method, terminating when the optimality gap falls below a predefined tolerance, tol. In this latter scenario, the algorithm can achieve early termination, thereby reducing the computational time associated with the final, most complex search iterations.

While evaluating candidate splits, the algorithm invokes the BR function, applying it to subtrees with a depth parameter of $d - 1$. This recursive application persists until the base case, $d = 1$, is reached. This method is a global optimal method, as proved later.

*Remark* B.1. The reduction strategy contributes the most to the efficiency in this algorithm (see Section A.6.2), with a trivial subtraction operation. It is similar to the first lower bounding problem proposed by Mazumder et al. (2022) but is generalized to any depth. The other lower bounding problems proposed by Hua et al. (2022); Mazumder et al. (2022) involve solving complicated lower bounding problems, leading to additional computational overhead. In contrast, BR requires only the computation of $\delta(\bar{k})$.

### B.3 PROOF OF THEOREM 2.1

Define $L(\mathcal{I})$ as the optimal loss of the dataset indexed by $\mathcal{I}$ when training a DT of a fixed depth:

$$L(\mathcal{I}) = \min_{T \in \mathcal{T}} L(\mathcal{I}, T). \tag{11}$$

We can give a boundary analysis of the loss function $L$ in Equation (1):

**Lemma B.2.** *For two sample index set $\mathcal{I}_1$ and $\mathcal{I}_2$ with $\mathcal{I}_1 \subseteq \mathcal{I}_2$, let $n_1$ and $n_2$ be the element numbers of $\mathcal{I}_1$ and $\mathcal{I}_2$, where $n_2 \geq n_1$, we have*

$$0 \leq L(\mathcal{I}_2) - L(\mathcal{I}_1) \leq n_2 - n_1.$$

*Proof.* From the formulation of Equation (1), we have:

$$L(\mathcal{I}_1) = \min_{T \in \mathcal{T}} \sum_{i \in \mathcal{I}_1} \ell(y_i, T(x_i)) \leq \min_{T \in \mathcal{T}} \sum_{i \in \mathcal{I}_1 \cup \{\mathcal{I}_2 \setminus \mathcal{I}_1\}} \ell(y_i, T(x_i))$$

$$= \min_{T \in \mathcal{T}} \sum_{i \in \mathcal{I}_2} \ell(y_i, T(x_i)) = L(\mathcal{I}_2). \tag{12}$$

So, we have $L(\mathcal{I}_1) \leq L(\mathcal{I}_2)$, which implies that prediction errors monotonically increase with sample numbers. Suppose the optimal tree for $\mathcal{I}_1$ is $T_1^*$, and for $\mathcal{I}_2$ we have

$$L(\mathcal{I}_2) \leq L(\mathcal{I}_2, T_1^*) = \sum_{i \in \mathcal{I}_2} \ell(y_i, T_1^*(x_i))$$

$$= \sum_{i \in \mathcal{I}_1} \ell(y_i, T_1^*(x_i)) + \sum_{i \in \mathcal{I}_2 \setminus \mathcal{I}_1} \ell(y_i, T_1^*(x_i))$$

$$= L(\mathcal{I}_1) + \sum_{i \in \mathcal{I}_2 \setminus \mathcal{I}_1} \ell(y_i, T_1^*(x_i)), \tag{13}$$

where the first inequality is because $L(\mathcal{I}_2)$ is the optimal value. Then we get:

$$L(\mathcal{I}_2) - L(\mathcal{I}_1) \leq \sum_{i \in \mathcal{I}_2 \setminus \mathcal{I}_1} \ell(y_i, T_1^*(x_i)) \leq \sum_{i \in \mathcal{I}_2 \setminus \mathcal{I}_1} \mathbf{1} = n_2 - n_1. \tag{14}$$

Now we have $0 \leq L(\mathcal{I}_2) - L(\mathcal{I}_1) \leq n_2 - n_1$, which bounds the loss function $L$. $\square$

**Theorem 2.1.** *For any $\mathcal{I} \subseteq [n]$ and $d \geq 1$, let $k_1$ and $k_2$ be two split candidates based on feature $a$. Then we have*

$$|V_d(\mathcal{I}, a, k_1) - V_d(\mathcal{I}, a, k_2)| \leq |\zeta(k_1) - \zeta(k_2)|. \tag{15}$$

*Proof.* Suppose $0 \leq k_1 \leq k_2 \leq n_a$, and we define $L(\mathcal{I})$ as Equation (11) in Theorem B.2. From the Equation (3), we have

$$V_d(\mathcal{I}, a, k_1) = L(\mathcal{I}_{[0,k_1]}^a) + L(\mathcal{I}_{[k_1,n_a]}^a), \tag{16}$$

$$V_d(\mathcal{I}, a, k_2) = L(\mathcal{I}_{[0,k_2]}^a) + L(\mathcal{I}_{[k_2,n_a]}^a). \tag{17}$$

According to Theorem B.2, we have

$$V_d(\mathcal{I}, a, k_2) - V_d(\mathcal{I}, a, k_1) = L(\mathcal{I}_{[0,k_2]}^a) + L(\mathcal{I}_{[k_2,n_a]}^a) - L(\mathcal{I}_{[0,k_1]}^a) - L(\mathcal{I}_{[k_1,n_a]}^a)$$

$$= \{L(\mathcal{I}_{[0,k_2]}^a) - L(\mathcal{I}_{[0,k_1]}^a)\} - \{L(\mathcal{I}_{[k_1,n_a]}^a) - L(\mathcal{I}_{[k_2,n_a]}^a)\}$$

$$\leq \zeta(k_2) - \zeta(k_1). \tag{18}$$

We take the case where $0 \leq k_2 \leq k_1 \leq n_a$ into consideration, which can be easily proven by reversal. Consequently, we get $|V_d(\mathcal{I}, a, k_1) - V_d(\mathcal{I}, a, k_2)| \leq |\zeta(k_1) - \zeta(k_2)|$. $\square$

### B.4 PROOF OF THEOREM 2.2

**Theorem 2.2.** *For any $\mathcal{I} \subseteq [n]$ and $d \geq 1$, let $V_d(\mathcal{I}, a, \bar{k})$ be a loss value of Equation (3) corresponding to the split rule $\beta_{\bar{k}}^a$, and an upper bound $U$ (assume $U \leq V_d(\mathcal{I}, a, \bar{k})$) of this problem, as well as the historical best loss in the searching process. Define*

$$\delta(\bar{k}) := V_d(\mathcal{I}, a, \bar{k}) - U. \tag{19}$$

*Then, for any $|\zeta(\bar{k}) - \zeta(k)| \leq \delta(\bar{k})$, we have $V_d(\mathcal{I}, a, k) \geq U$, namely, the optimal split index is not in*

$$\Delta = \{k \mid |\zeta(\bar{k}) - \zeta(k)| \leq \delta(\bar{k}), \ k \in \mathcal{K}_i\}. \tag{20}$$

*Proof.* Suppose there is a split $k' \in \Delta$ such that $V_d(\mathcal{I}, a, k') < U$. According to Theorem 2.1, we have

$$|V_d(\mathcal{I}, a, \bar{k}) - V_d(\mathcal{I}, a, k')| \leq |\zeta(\bar{k}) - \zeta(k')| \leq \delta = V_d(\mathcal{I}, a, \bar{k}) - U. \tag{21}$$

Since $V_d(\mathcal{I}, a, k') < U \leq V_d(\mathcal{I}, a, \bar{k})$, we have

$$V_d(\mathcal{I}, a, \bar{k}) - V_d(\mathcal{I}, a, k') > V_d(\mathcal{I}, a, \bar{k}) - U > 0, \tag{22}$$

which is a contradiction of Equation (21). So there does not exist a split in $\Delta$ with a loss value less than $U$, i.e., the optimal solution is not in $\Delta$. $\square$

### B.5 OPTIMALITY PROOF OF ALGORITHM 4

**Theorem B.3.** *Algorithm 4 converges to the global optimum of Equation (3).*

*Proof.* Since Equation (3) is a bilevel optimization problem, we analyze both *ULP* and *LLP*. We begin with the case of a depth-1 tree (including R-CART), where the *LLP* corresponds to optimizing a constant fit. To establish the optimality of Algorithm 4 for a depth-1 tree, it suffices to show that the reduction strategy does not exclude any optimal solution, as the algorithm without reduction effectively performs an exhaustive enumeration. Theorem 2.2 demonstrates that the optimal split is never removed from the reduced set $\Delta$, thereby ensuring that Algorithm 4 converges to the optimal solution of the *LLP*.

Next, we assume that Algorithm 4 converges to the global optimal solution for a depth-$(d-1)$ tree. For a depth-$d$ tree, each *LLP* subproblem corresponds to optimizing a depth-$(d-1)$ tree, which, by assumption, is solved to global optimality. The *ULP* follows the same proof structure as the depth-1 case. Consequently, by induction, Algorithm 4 converges to the global optimum of Equation (3). $\square$

## C ANALYSIS ABOUT THE APPROXIMATE METHOD

### C.1 RECALL THE BASICS OF CART

Before analyzing the ABR method, we recall some basics of CART. We first analyze some properties of the DT model. Let $\mathcal{I}_t$ denote the data in the node $t \in \mathcal{N}_B \cup \mathcal{N}_L$, then the potential loss of this node can be calculated as:

$$V_0(\mathcal{I}_t) = \min_{c \in \mathcal{N}_c} L(\mathcal{I}_t, c) \tag{23}$$

The parameters of this node involve minimizing the combined loss of the potential children:

$$a', k' = \arg \min_{a \in [p], k \in \mathcal{K}^a} V_0(\mathcal{I}_{t,[0,k]}^a) + V_0(\mathcal{I}_{t,[k,n_a]}^a) \tag{24}$$

where $\mathcal{I}_{t,[0,k]}^a$ and $\mathcal{I}_{t,[k,n_a]}^a$ respectively represent the data subsets assigned to the left and right child nodes resulting from the split of the data $\mathcal{I}_t$ based on $(a, k)$.

**Depth-1 optimality** For a sample subset $\mathcal{I} \subseteq [n]$, a depth-1 DT comprises two optimal constant fits, i.e., two optimal depth-0 subtrees, expressed by $V_0(\mathcal{I}) = \min_{c \in \mathcal{N}_c} L(\mathcal{I}, c)$, which denotes the loss of the best constant approximation to $\mathcal{Y}$, in the same manner as CART. For $d = 1$, CART solves a one-stage decision problem, which, according to the greedy nature, is optimal.

**Monotonicity** The loss function of `CART` exhibits monotonicity with respect to tree depth, such that $W_d(\mathcal{I}) \leq W_{d-1}(\mathcal{I})$. This is attributed to the monotonic nature of the branching operation concerning the count of correctly classified samples (accuracy). Branching will either increase this count (thereby decreasing the loss) or, in the worst-case scenario where the newly formed leaves offer no improvement over their parent node, the count will remain unchanged.

For example, we consider a node $t$ with data $\mathcal{I}_t$, the potential loss is $V_0(\mathcal{I}_t)$ as defined by Equation (23), with prediction $c_1$. Then, we branch the node to obtain two branch nodes $\mathcal{I}_{2t}, \mathcal{I}_{2t+1}$, where $\mathcal{I}_{2t} \cup \mathcal{I}_{2t+1} = \mathcal{I}_t$. The losses of these new nodes are $V_0(\mathcal{I}_{2t})$ and $V_0(\mathcal{I}_{2t+1})$ with predictions $c_2$ and $c_3$, respectively. Then we have:

$$
\begin{aligned}
V_0(\mathcal{I}_t) = \sum_{i \in \mathcal{I}} \mathbb{1}\{y_i \neq c_1\} &= \sum_{i \in \mathcal{I}_{2t}} \mathbb{1}\{y_i \neq c_1\} + \sum_{i \in \mathcal{I}_{2t+1}} \mathbb{1}\{y_i \neq c_1\} \\
&\geq \sum_{i \in \mathcal{I}_{2t}} \mathbb{1}\{y_i \neq c_2\} + \sum_{i \in \mathcal{I}_{2t+1}} \mathbb{1}\{y_i \neq c_3\} \\
&= V_0(\mathcal{I}_{2t}) + V_0(\mathcal{I}_{2t+1}),
\end{aligned}
\tag{25}
$$

The inequality holds because $c_2$ and $c_3$ are chosen as the predictions that minimize the misclassifications within their respective nodes $2t$ and $2t+1$ (defined as the most frequent class in Equation (23)). By definition, the optimal prediction $c_2$ for node $\mathcal{I}_{2t}$ must classify at least as many samples correctly within $\mathcal{I}_{2t}$ as any other prediction, including $c_1$. Thus, $\sum_{i \in \mathcal{I}_{2t}} \mathbb{1}\{y_i \neq c_2\} \leq \sum_{i \in \mathcal{I}_{2t}} \mathbb{1}\{y_i \neq c_1\}$, and similarly for $c_3$ over $\mathcal{I}_{2t+1}$. Therefore, the loss after branching is less than or equal to the value before branching.

### C.2 A REDUCED `CART` METHOD

The BR method has a small application when $d = 1$, as we have claimed in Algorithm 5 outlines the steps of a reduced `CART` within the bilevel framework, which is applied to MHABR to solve $\text{ABR}(\mathcal{I}, 1)$. The reduction strategy in Algorithm 4 is also applied to `CART`. The reduced `CART` is named as `R-CART`, which is similar to a special case of Algorithm 4 to depth-1 trees. Since MHABR

---

**Algorithm 5** A reduced `CART`

1: **function** `R-CART`($\mathcal{I}$):
2:    **for** $a \in [p]$ **do**
3:       Sort $\mathcal{I}$ to obtain $\mathcal{I}^a, \mathcal{K}^a$, and set $\mathbb{K} \leftarrow \{\mathcal{K}^a\}$;
4:       **for** $\bar{k} \in \mathbb{K}$ **do**
5:          Select $\bar{k} \leftarrow \mathbb{K}[1]$, obtain $\mathcal{I}^a_{[0,\bar{k}]}, \mathcal{I}^a_{[\bar{k},n_a]}$;         # *select the first element*
6:          Compute $V_0(\mathcal{I}^a_{[0,\bar{k}]}) \leftarrow \min_c L(\mathcal{I}^a_{[0,\bar{k}]}, c)$, and $c_1 \leftarrow \arg\min_c L(\mathcal{I}^a_{[0,\bar{k}]}, c)$;
7:          Compute $V_0(\mathcal{I}^a_{[\bar{k},n_a]}) \leftarrow \min_c L(\mathcal{I}^a_{[\bar{k},n_a]}, c)$, and $c_2 \leftarrow \arg\min_c L(\mathcal{I}^a_{[\bar{k},n_a]}, c)$;
8:          Calculate $V_1(\mathcal{I}, a, \bar{k}) = V_0(\mathcal{I}^a_{[0,\bar{k}]}) + V_0(\mathcal{I}^a_{[\bar{k},n_a]})$;
9:          **if** $V_d(\mathcal{I}, a, \bar{k}) < U$ **then**
10:            Update $U \leftarrow V_d(\mathcal{I}, a, \bar{k})$, $T \leftarrow [a, \bar{k}, c_1, c_2]$;
11:          Compute $\delta(\bar{k}) \leftarrow \max\{V_d(\mathcal{I}, a, \bar{k}) - U, 1\}$;     # *at least reduce $\bar{k}$, so $\delta(\bar{k}) \geq 1$*
12:          $\mathbb{K} \leftarrow \{k \mid k \geq \texttt{searchfirst}(\bar{k} + \delta(\bar{k}), \mathbb{K}), k \in \mathbb{K}\}$;   # *jump the reduced and repeated elements*
13: **return:** Optimal loss $U$ and decision tree $T$.

---

involves multiple uses of `CART`, even a modest improvement in `CART` would significantly boost efficiency and have a cumulative effect on the overall algorithm. The reduction strategy provides a more efficient gain for $d = 2$ compared to $d = 1$, as the variability in subtree loss increases with depth. In the absence of reduction, Algorithm 4 converges to the global optimum of Equation (3) within at most $\sum_{a=1}^{p} n_a$ evaluations of `CART`. Unlike our BR method, `CART` follows a forward search approach rather than using a bisection branching strategy. Notably, the `R-CART` algorithm can be viewed as a special case of the BR method proposed in Algorithm 4, corresponding to a depth-1 tree without recursion for lower depths.

## C.3 Analysis of moving-horizon improvements based on ABR

The core of the moving-horizon improvement lies in ABR achieving better accuracy than `CART`, particularly in solving *LLP*s. We first prove Theorem 3.1. In this subsection, we only consider the original optimization problem that ABR solves, instead of the algorithm (we will discuss in the next subsection).

**Theorem 3.1** *Suppose there is no reduction strategy in Algorithm 2, let $W_d(\mathcal{I})$ and $\hat{V}_d(\mathcal{I})$ be defined as above, it holds that $\hat{V}_d(\mathcal{I}) \leq W_d(\mathcal{I})$ for all $d > 0$.*

*Proof.* Since there is no reduction, we consider a complete search of the range. From Equation (8), $\hat{V}_d(\mathcal{I})$ is calculated by solving

$$
\begin{aligned}
\hat{V}_d(\mathcal{I}) &= \min_{a \in [p], k \in \mathcal{K}^a} \hat{V}_d(\mathcal{I}, a, k) \\
&= \min_{a \in [p], k \in \mathcal{K}^a} \{ L(\mathcal{I}^a_{[0,k]}, \hat{T}_L) + L(\mathcal{I}^a_{[k,n_a]}, \hat{T}_R) \} \\
&= \min_{a \in [p], k \in \mathcal{K}^a} \{ W_{d-1}(\mathcal{I}^a_{[0,k]}) + W_{d-1}(\mathcal{I}^a_{[k,n_a]}) \}
\end{aligned}
\tag{26}
$$

and $W_d(\mathcal{I})$ can be calculated as

$$
W_d(\mathcal{I}) = \hat{V}_d(\mathcal{I}, a', k') = W_{d-1}(\mathcal{I}^{a'}_{[0,k']}) + W_{d-1}(\mathcal{I}^{a'}_{[k',n_{a'}]}),
\tag{27}
$$

where $(a', k')$ denotes the root node parameters obtained by `CART` (as well as an optimal depth-1 tree), and is calculated by:

$$
\begin{aligned}
a', k' &= \arg \min_{a \in [p], k \in \mathcal{K}^a} \left\{ V_0(\mathcal{I}^a_{[0,k]}) + V_0(\mathcal{I}^a_{[k,n_a]}) - V_0(\mathcal{I}) \right\} \\
&= \arg \min_{a \in [p], k \in \mathcal{K}^a} \left\{ V_0(\mathcal{I}^a_{[0,k]}) + V_0(\mathcal{I}^a_{[k,n_a]}) \right\} \\
&= \arg \min_{a \in [p], k \in \mathcal{K}^a} \left\{ \min_{c_1 \in \mathcal{N}_c} L(\mathcal{I}^a_{[0,k]}, c_1) + \min_{c_2 \in \mathcal{N}_c} L(\mathcal{I}^a_{[k,n_a]}, c_2) \right\}
\end{aligned}
$$

where $V_0(\mathcal{I}) = \min_{c \in \mathcal{N}_c} L(\mathcal{I}, c)$ can be easily calculated as $\mathcal{I}$ is given, and it represents the rate of the most frequent class. So $L(\mathcal{I}, (a', k')) = W_1(\mathcal{I})$ holds. It is obvious that $(a', k')$ is a feasible solution in the range of Equation (26), while $\hat{V}_d(\mathcal{I})$ is the optimal value of Equation (26). Then we have $\hat{V}_d(\mathcal{I}) \leq W_d(\mathcal{I})$ for all $a \in [p], k \in \mathcal{K}^a$. So $\hat{V}_d(\mathcal{I}) \leq W_d(\mathcal{I})$, this lemma is proved. □

Theorem 3.1 proves an improvement of the approximate method compared to `CART`. It should be noted that `CART` is included to approximately evaluate the lower-level subtrees, meaning that for the subtrees, Theorem 3.1 will result in better accuracy. Suppose the refined subtree has at least an improvement of $\eta$ for any decision tree deeper than depth 2, then MH improves at least $(2^{d-1} - 1)\eta$ of the loss for a DT with depth $d$. Now, we discuss this improvement.

From the steps of `CART` algorithm, $W_d(\mathcal{I})$ can be calculated as

$$
W_d(\mathcal{I}) = W_{d-1}(\mathcal{I}^{a'}_{[0,k']}) + W_{d-1}(\mathcal{I}^{a'}_{[k',n_{a'}]}),
\tag{28}
$$

$$
\hat{V}_d(\mathcal{I}) = W_{d-1}(\mathcal{I}^{a^*}_{[0,k^*]}) + W_{d-1}(\mathcal{I}^{a^*}_{[k^*,n_{a^*}]}),
\tag{29}
$$

where $(a', k')$ and $(a^*, k^*)$ are the root node parameters obtained by `CART` and approximate method respectively. By Theorem 3.1, we have $\hat{V}_d(\mathcal{I}) \leq W_d(\mathcal{I})$ for any $d \geq 2$. Then $\hat{V}_{d-1}(\mathcal{I}) \leq W_{d-1}(\mathcal{I})$, meaning that

$$
\hat{V}_d(\mathcal{I}) = W_{d-1}(\mathcal{I}^{a^*}_{[0,k^*]}) + W_{d-1}(\mathcal{I}^{a^*}_{[k^*,n_{a^*}]}) \geq \hat{V}_{d-1}(\mathcal{I}^{a^*}_{[0,k^*]}) + \hat{V}_{d-1}(\mathcal{I}^{a^*}_{[k^*,n_{a^*}]}).
\tag{30}
$$

So we iteratively have

$$
\hat{V}_{d-1}(\mathcal{I}) \geq \hat{V}_{d-2}(\mathcal{I}^{a^*}_{[0,k^*]}) + \hat{V}_{d-2}(\mathcal{I}^{a^*}_{[k^*,n_{a^*}]}),
$$

$$
\hat{V}_{d-2}(\mathcal{I}) \geq \hat{V}_{d-3}(\mathcal{I}^{a^*}_{[0,k^*]}) + \hat{V}_{d-3}(\mathcal{I}^{a^*}_{[k^*,n_{a^*}]}),
$$

$$
\cdots
$$

until

$$\hat{V}_2(\mathcal{I}) = \hat{V}_1(\mathcal{I}^{a^*}_{[0,k^*]}) + \hat{V}_1(\mathcal{I}^{a^*}_{[k^*,n_{a^*}]}) = W_1(\mathcal{I}^{a^*}_{[0,k^*]}) + W_1(\mathcal{I}^{a^*}_{[k^*,n_{a^*}]}). \tag{31}$$

Thus, the MH process guarantees an improvement in the accuracy of DTs deeper than 2. In addition, if we suppose there is a positive $\eta$ such that $\eta \leq W_d(\mathcal{I}) - \hat{V}_d(\mathcal{I})$ for any $d \geq 2$, then MH improves at least $(2^{d-1} - 1)\eta$ of the loss.

## C.4 DISCUSSION OF THE REDUCTION STRATEGY IN THE APPROXIMATE METHOD

The information obtained from Theorem 2.1 can be regarded as a lower bound of $V_d(\mathcal{I}, a, k)$, using a calculated optimal loss and the upper bound, and maintaining accuracy when the loss is globally optimal. However, Theorem 2.1 holds only if the loss is globally optimal. Therefore, the real lower bound is different. Now we analyze the reduction strategy of the approximate method.

In Algorithm 3, the approximation considers that $\hat{V}_d(\mathcal{I}, a, k) \approx V_d(\mathcal{I}, a, k)$, meaning that there is an error in the process. Suppose that $\hat{V}_d(\mathcal{I}, a, k) - V_d(\mathcal{I}, a, k) = \lambda(k)$, where $\hat{V}_d(\mathcal{I}, a, k) \leq V_d(\mathcal{I}, a, k)$ because $V_d(\mathcal{I}, a, k)$ is the optimal value. Then, for a given loss of split $\bar{k}$, we have the following:

$$\hat{V}_d(\mathcal{I}, a, k) \geq V_d(\mathcal{I}, a, \bar{k}) + \lambda(k) - |\zeta(\bar{k}) - \zeta(k)|,$$
$$V_d(\mathcal{I}, a, \bar{k}) = \hat{V}_d(\mathcal{I}, a, \bar{k}) - \lambda(\bar{k}).$$

Then the lower bound of $V_d(\mathcal{I}, a, k)$ can be calculated as

$$\hat{V}_d(\mathcal{I}, a, k) \geq \hat{V}_d(\mathcal{I}, a, \bar{k}) - |\lambda(\bar{k}) - \lambda(k)| - |\zeta(\bar{k}) - \zeta(k)|. \tag{32}$$

However, $|\lambda(\bar{k}) - \lambda(k)|$ is generally challenging to evaluate. Since $\hat{V}_d(\mathcal{I}, a, k)$ does not aim for the global optimal solution, this error can be tolerated. As mentioned in Section 2, Equation (7) offers a highly efficient, somewhat loose, lower bound.

The error $|\lambda(\bar{k}) - \lambda(k)|$ and the gap between $|\zeta(\bar{k}) - \zeta(k)|$ and $V_d(\mathcal{I}, a, k)$ often offset each other. Therefore, Algorithm 3 slightly compromises accuracy to ensure high efficiency, bringing stochastics, but the MH process can further refine the accuracy.

We now turn to a comparison with `CART` from a bilevel perspective. We know that the loss obtained by `CART` can be expressed as:

$$\mathcal{L}(W_d, \Omega) := W_d(\mathcal{I}) = W_{d-1}(\mathcal{I}^{a'}_{[0,k']}) + W_{d-1}(\mathcal{I}^{a'}_{[k',n_{a'}]}) \tag{33}$$
$$\text{where } a', k' = \arg\min_{\Omega} W_0(\mathcal{I}^a_{[0,k]}) + W_0(\mathcal{I}^a_{[k,n_a]})$$

where the equation is obtained from Equation (24) and because $V_0(\mathcal{I}) = W_0(\mathcal{I})$. The solution obtained by MHABR can be obtained from Equation (8), considering the reduction here, denoted as

$$\mathcal{L}(\hat{V}_d, \hat{\Omega}) : \hat{V}_d(\mathcal{I}) = W_{d-1}(\mathcal{I}^{\hat{a}}_{[0,\hat{k}]}) + W_{d-1}(\mathcal{I}^{\hat{a}}_{[\hat{k},n_{\hat{a}}]}) \tag{34}$$
$$\text{where } \hat{a}, \hat{k} = \arg\min_{\hat{\Omega}} W_{d-1}(\mathcal{I}^a_{[0,k]}) + W_{d-1}(\mathcal{I}^a_{[k,n_a]})$$

where $\hat{\Omega} \subseteq \Omega$ denotes the reduced feasible range. It is obvious that $\mathcal{L}(\hat{V}_d, \hat{\Omega}) \leq \mathcal{L}(W_d, \Omega)$ from the definition of $\hat{V}_d$. Let $\hat{\Omega}^- = \Omega \setminus \hat{\Omega}$, then the comparison is about $\mathcal{L}(\hat{V}_d, \hat{\Omega})$ and $\mathcal{L}(W_d, \hat{\Omega}^-)$.

## C.5 COMPUTATIONAL COST ANALYSIS

Now, we are discussing the computational cost of the algorithms proposed in this paper.

**Theorem 3.2** *Suppose there are at most $\tilde{n}$ split candidates at each layer, then the cost of `CART` is bounded by $\mathcal{O}(\tilde{n}d)$, BR is bounded by $\mathcal{O}(\tilde{n}^d)$, and `MHABR` is bounded by $\mathcal{O}(\tilde{n}^2 \frac{d(d-1)}{2})$ for $d \geq 2$.*

*Proof.* Given a maximum of $\tilde{n}$ split candidates at each layer, the greedy `CART` algorithm determines tree parameters node by node, as detailed in Section C.1. Consequently, the computational cost of `CART` at each layer is independent of other layers, and the aggregate computational complexity can be expressed as $\mathcal{O}(\tilde{n}d)$.

Subsequently, the BR method, being a global optimization approach, recursively solves for the subtrees until a depth of 1 is reached. Therefore, its computational cost is determined by the product of the cost at each of the d layers, yielding a complexity of $\mathcal{O}(\tilde{n}) \times \cdots \times \mathcal{O}(\tilde{n}) = \mathcal{O}(\tilde{n}^d)$.

Then, based on the costs of CART and BR, we can obtain the cost of ABR. For the *LLP*, there are left and right subtrees solved by CART. We suppose that at most $\tilde{n}_L$ and $\tilde{n}_R$ split candidates for the left and right subtrees, respectively. Then we have $\tilde{n}_L + \tilde{n}_R \leq \tilde{n}$. The subtrees have depth $d-1$, the cost is $\mathcal{O}(\tilde{n}_L(d-1)) + \mathcal{O}(\tilde{n}_R(d-1)) \leq \mathcal{O}(\tilde{n}(d-1))$. Then for the *ULP*, the cost is still bounded by $\tilde{n}$. So, the total cost of ABR is bounded by $\mathcal{O}(\tilde{n}^2(d-1))$ for $d \geq 2$.

At last, we consider the cost of MH. For each MH iteration with the depth $d_{sub}$, the cost is $\mathcal{O}(\tilde{n}^2(d_{sub}-1))$, and MH runs at most $2^{d-1}-1$ iterations. So the cost of MHABR is calculated as

$$
\begin{aligned}
& \tilde{n}^2(d-1) + \sum_{i=2}^{2} \tilde{n}_i^2(d-2) + \cdots + \sum_{i=2^{d-2}}^{2^{d-1}} \tilde{n}_i^2 \\
& \leq \tilde{n}^2(d-1) + (\sum_{i=1}^{2} \tilde{n}_i)^2(d-2) + \cdots + (\sum_{i=2^{d-2}}^{2^{d-1}} \tilde{n}_i)^2 \\
& \leq \tilde{n}^2(d-1) + \tilde{n}^2(d-2) + \cdots + \tilde{n}^2 \\
& = \tilde{n}^2 \frac{d^2-d}{2}
\end{aligned}
\tag{35}
$$

Therefore, we can conclude that the cost of MHABR is bounded by $\mathcal{O}(\tilde{n}^2 \frac{d(d-1)}{2})$ for $d \geq 2$. Note that, when $d = 1$, all the methods are bounded by $\mathcal{O}(\tilde{n})$. $\square$

# D  DETAILS OF NUMERICAL EXPERIMENTS

## D.1  INFORMATION OF SMALL-SCALE DATASETS

For the 51 small datasets, we do not introduce the parameters $\varepsilon$ and $\theta$. The details of these datasets, along with the results of the algorithms discussed in Section 4.1, are presented below.

## D.2  IMPLEMENTATION DETAILS

Details and experimental settings of all comparison algorithms are stated below. Unless specified, implementations of algorithms used in our experiments are obtained from their original authors.

**MHABR**: Our algorithm is implemented in `Julia`. For the default version, we adopt the reduction strategy with parameters $\varepsilon = 0$ and $\theta = 1$. To construct MHABR$^2$, which enlarges the *ULP* to include the top two layers of nodes, we modify the evaluation function in ABR recursion as follows:

$$
\text{MHABR}: \begin{cases} \text{BR}, & d - d_{sub} \leq 1 \\ \text{CART}, & \text{otherwise} \end{cases} \quad \Rightarrow \quad \text{MHABR}^2: \begin{cases} \text{BR}, & d - d_{sub} \leq 2 \\ \text{CART}, & \text{otherwise.} \end{cases}
$$

**CART** (Sadeghi et al., 2022):  We use the implementation from the `Julia` package `DecisionTree`, selecting entropy loss as it provides the best results among the three options; its performance may occasionally surpass MHABR and DPDT on several datasets.

**DPDT** (Kohler et al., 2025): This algorithm is written in Python and calls CART by `Python` package `sklearn`, using the **Gini loss** (default). Therefore, in our results, it may be surpassed by CART.

**LS-OCT** (Dunn, 2018): Since the original code is not available, we implement both methods in `Julia` and call `Gurobi` to solve MIP models.

**DL8.5** (Aglin et al., 2020): This algorithm is written in `C++` and is run as an extension of `Python`. The current version also integrates the methods of `MurTree` (Demirović et al., 2022). Besides, we utilize the `GUESS` (McTavish et al., 2022) for binarization, which has the best performance.

**Quant-BnB** (Mazumder et al., 2022): This algorithm is written in `Julia`.

Table 11: The information of 51 small-scale datasets.

| Dataset Name | $n$ | $p$ | Class | Dataset Name | $n$ | $p$ | Class |
|---|---|---|---|---|---|---|---|
| Soybean-small | 47 | 35 | 4 | Body | 507 | 5 | 2 |
| Echocardiogram | 61 | 11 | 2 | Climate-model-crashes | 540 | 20 | 2 |
| Hepatitis | 80 | 19 | 2 | Monks-problems-3 | 554 | 6 | 2 |
| Fertility | 100 | 9 | 2 | Monks-problems-1 | 556 | 6 | 2 |
| Acute-inflammations-1 | 120 | 6 | 2 | Breast-cancer-diagnosti | 569 | 30 | 2 |
| Acute-inflammations-2 | 120 | 6 | 2 | Monks-problems-2 | 600 | 6 | 2 |
| Hayes–roth | 132 | 5 | 3 | Balance-scale | 625 | 4 | 3 |
| Iris | 150 | 4 | 3 | Credit-approval | 653 | 15 | 2 |
| Teaching-assistant-evaluation | 151 | 5 | 3 | Breast-cancer | 683 | 9 | 2 |
| Wine | 178 | 13 | 3 | Blood-transfusion | 748 | 4 | 2 |
| Breast-cancer-prognostic | 194 | 31 | 2 | Mammographic-mass | 830 | 5 | 2 |
| Parkinsons | 195 | 23 | 2 | Tic-tac-toe-endgame | 958 | 9 | 2 |
| Connectionist-bench-sonar | 208 | 60 | 2 | Connectionist-bench | 990 | 13 | 11 |
| Image-segmentation | 210 | 19 | 7 | Statlog-project-German-credit | 1,000 | 20 | 2 |
| Seeds | 210 | 7 | 3 | Concrete | 1,030 | 8 | 3 |
| Glass | 214 | 9 | 6 | Banknote-authentication | 1,372 | 4 | 2 |
| Thyroid-disease-new-thyroid | 215 | 5 | 3 | Contraceptive-method-choice | 1,473 | 9 | 3 |
| Congressional-voting-records | 232 | 16 | 2 | Car-evaluation | 1,728 | 6 | 4 |
| Spect-heart | 267 | 22 | 2 | Ozone-level-detection-eight | 1,847 | 72 | 2 |
| Spectf-heart | 267 | 44 | 2 | Ozone-level-detection-one | 1,848 | 72 | 2 |
| Cylinder-bands | 277 | 39 | 2 | Seismic-bumps | 2,584 | 18 | 2 |
| Heart-disease-Cleveland | 282 | 13 | 5 | Chess-king-rook-versus-king-pawn | 3,196 | 36 | 2 |
| Haberman-survival | 306 | 3 | 2 | Thyroidann | 3,772 | 21 | 3 |
| Ionosphere | 351 | 34 | 2 | Wall-following-robot-2 | 5,456 | 2 | 4 |
| Dermatology | 358 | 34 | 6 | Thyroid-disease-ann-thyroid | 7,200 | 21 | 3 |
| Thoracic-surgery | 470 | 16 | 2 | | | | |

**IAI-OCT** (Interpretable AI, 2025): This method is a commercial package based on the formulation of Bertsimas & Dunn (2017), and is integrated (forced) with a grid search for hyperparameter tuning.

Additionally, we experimented with **STreeD** (van der Linden et al., 2023), implemented in C++ with a `Python` interface. However, due to changes in computational hardware, a fair comparison could not be ensured; therefore, we omit its results.

### D.3 CONFIGURATION OF ADDITIONAL TECHNIQUES FOR LARGE-SCALE DATASETS

The results of these datasets with a sample size greater than $10,000$ are given as follows. For the 3 medium datasets, we do not introduce the parameters $\varepsilon$ and $\theta$; MHABR successfully completes all the training tasks. For the three large-scale datasets, we configure the algorithms to terminate within the time limit. The configurations are SUSY: $\varepsilon = 0.01, \theta = 0.25$, HIGGS: $\varepsilon = 0.01, \theta = 0.5$, WESAD: $\varepsilon = 0.00025, \theta = 0.25$ respectively.

## DECLARATION OF LLM USAGE

In the preparation of this manuscript, LLM is used exclusively for writing, editing, and formatting assistance. The use of the LLM is limited to improving clarity, grammar, and presentation, and it did not influence any of the core methodology, experimental design, analysis, or the scientific rigor of this work. All technical content, results, and interpretations are the original contributions of the authors.

