# OpenReview forum: "Beyond Greedy: Towards Optimal Deep Classification Trees"
_ICLR.cc/2026/Conference — Submitted to ICLR 2026_

### Official Review · Reviewer_nJuV · 2025-10-27

**Soundness:** 4
**Presentation:** 1
**Contribution:** 3
**Rating:** 4
**Confidence:** 5

**Summary:**

Authors propose a new heuristic decision tree induction algorithm supposidly bridging the gap between optimality and scalability.
The proposed MHABR iteratively refines an ABR global tree by calling ABR on sub-trees of the global tree. ABR computes a tree with a root node that is optimal w.r.t. its induced greedy sub-trees (obtained with CART). The idea of refinining an ABR tree is good. Performance guarantees and numerical performances are OK. The key weaknesses that justify my score are the lack of clarity of some section and some very subjective or wrong tables and figures, and above all the lack ok positioning w.r.t to the closest related work (DPDT).
I would accept the paper if a core part of the experiments focuses on the optimality/runtime pareto front of DPDT and MHABR . Similarly you can also do the test accuracy/runtime pareto front of DPDT and MHABR. You can do so for a fixed large datasets when inducing trees of different depth. The idea of MHABR is good but authors ought to get their experiments to the highest standards of excellence (including hyperparameters search for example).

**Strengths:**

## The paper is well written
The introduction is clear and presents the problem clearly,
There is (almost) no missing related work except Blanc, et. al. Harnessing the power of choice in NeurIPS 2024, and Lookahead and Pathology in Decision Tree Induction. Murthy 1995.
Figure 1 is great.

## Analysis
There is a nice theoritical guarantee that the proposed ABR is >= CART and is optimal for depth 2 trees (c.f. Lemma 3.1).
There is an analysis of the complexity of MHABR.

I would personally write the performances guarantees as theorems and the complexity as a proposition.

**Weaknesses:**

## Subjectivity or lack of rigour.
Table 1 is not very rigorous. First of all, of course that CART cannot deal with infinite depth or infinitely big datasets no physical system can. Still on table 1, the reported value for DPDT is incorrect: in the DPDT paper authors try a depth-10 decision tree on the KDD dataset (N= 5 million, p = 80) on table 3. In general, DPDT can be instantiated to be CART (from section 5.2 of the DPDT paper, last paragraph, DPDT with B=1 is CART, in python ```from dpdt import DPDTreeClassifier; clf = DPDTreeClassifier(cart_nodes_list(1,))``` is CART). Similarly, ABR, if the LLP problem is approximated from the root to the leaves, then ABR is CART. I would remove this table 1.

In Appendix A.5, figure 7 is not rigorous at all. For example, DPDT covers the whole pareto front of this subjective figure, and CART could also also returned the optimal tree if optimilaty is with respect to e.g. the depth-1 tree.

## Typos
Line 124: this sum only denotes the number of misclassif. if $l()$ is the 0-1 loss.
In general the trin induction problem is weirdly defined: in equation (1), V is an integer and not a tree. It would be better to have an argmin to show that you are look for a tree.
Line 149 about the root node is interesting but needs more explanations and/or a reference.

## Clarity
Second, section 3.3 is essentially null. The iterative refinement presented is not detailed: in figure 2, the schematics only shows that trees are recursive data structures but in no way does figure 2 nor sec. 3.3 present any new optimization problem nor algorithm.

The rest of section 3 helps *guess* what MHABR but it is very unclear: algorithms 1-3 are weird because algorithm 1, ABnR, is not defined in the text.

My educated guess, please correct me if I am wrong, is that MHABR is to run iteratively ABR on sub trees of a global ABR tree.
Supposidly, this should refine the global ABR tree but could also degrade it because there is no guarantee that change a subtree of the global ABR tree will not render, e.g, the global root node, less performent.

Despite the lack of clarity of section 3, I think the proposed MHABR is a good idea. It makes sense and the proof of theorem 3.2 is easy to follow: the complexity of MHABR is number of iteration of MH times complexity of ABR on the subtree.

## Better positioning w.r.t DPDT
In general, I think your work should better situate w.r.t to DPDT as DPDT also tackles "the urgent need for an advanced method that can deliver both high accuracy and scalability for deep decision trees on large-scale datasets." .

Up until section 3.3, the proposed ABR is exactly the baseline DPDT with hyperparameters B=Np at d < 2 and B = 1 for d >= 2 (Kohler et. al 2025 section 5.6). Similarly, ABR² is DPDT with B=Np at d < 2 and B = 1 for d >= 3.
To convince yourself of that you could run:
```python
from dpdt import DPDTreeClassifier
from sklearn.datasets import load_digits

DEPTH = 5 # can be changed
X, y = load_digits(return_X_y=True) # MNIST
N, p = len(X), len(X[0])
abr = DPDTreeClassifier(cart_nodes_list=(N * p, ), max_depth=DEPTH) # ABR
abr.fit(X, y) # this should return the same tree as ABR with a max depth of DEPTH
abr.score(X,y) # this should be the same optimality as ABR tree

abr2 = DPDTreeClassifier(cart_nodes_list=(N * p, N * p, ), max_depth=DEPTH) # ABR2
abr2.fit(X, y) # this should return the same tree as ABR2 with a max depth of DEPTH
abr2.score(X,y) # this should be the same optimality as ABR2 tree
```
The idea of line 276 of using CART to get approximated sub-trees for the LLP is from DPDT. Figure 1 is essentially the work of the DPDT paper. The real novely of the work starts from 3.4 onwards.

## Experiments
The main weakness for the experiment is that authors use default hyperparameters for most baselines.
Table 2 shows good train accuray at cheaper than optimal algs cost. This is essentially a property that bot CART and DPDT exhibits, with DPDT being the first algo that can really navigate the pareto front of optimality/scalability.
In appendix table 8, DPDT does not use parallel computation by default.
A more thorough comparison of CART vs DPDT vs MHABR ought to be in the main paper with hyperparameter search.

**Questions:**

- What do you mean by best outcome for alpha CART and alpha MHABR lines 390? Is it the best test accuracy or best train?
- Can you run, for depths 2 --> 20 on a large dataset CART(max_depth = d, random_state=42), DPDT(cart_nodes_list=(Np, ), max_depth=d, random_state=42) # ABR, MHABR(max_depth = d, random_state=42)? Then you can plot a pareto front of runtimes vs test/train scores.
- What are the techniques for light MHABR?

---

> ### Author Response · Authors · 2025-11-24
> **Reply to the comments (corresponding experiments are included in the revised paper).**
>
> Thank you for your valuable comments and insightful suggestions, which are extremely helpful in improving our paper. We fully agree with your recommendation to better position our study with respect to DPDT. We have incorporated this aspect and revised the paper accordingly. Our responses to your specific concerns are provided in detail below:
>
> ## Subjectivity or lack of rigour.
> Yes, we agree with you. Table 1 and figure 7 are too subjective and not very rigorous. We apologize for not being rigorous here. We have deleted table 1 and figure 7.
>
> ## Typo
> We apologize for the lack of clarity in the original text. We have now added an explicit 'argmin' equation to denote the optimization problem used to find the optimal tree. It was not a typographical error: we intended to use $V_d$ to represent the loss (value or misclassification). However, we agree that the notation was unclear, and we have revised this part of the manuscript for improved readability. $$T^*(\mathcal{I}) = \arg\min_{T\in\mathcal{T}_d} L(\mathcal{I},T)$$For line 149, we have added the references in the revised version.
> - Dwyer, Kenneth, and Robert Holte. "Decision tree instability and active learning." _European conference on machine learning_. Berlin, Heidelberg: Springer Berlin Heidelberg, 2007.
> - Shannon, William D., and David Banks. "Combining classification trees using MLE." _Statistics in medicine_ 18.6 (1999): 727-740.
>
> ## Clarity
> - Here, we want to illustrate the process of MH with an example, we have revised this part and re-plot a figure to illustrate the MH process, please see figure 2 in the revised paper for detail.
> - MHABR operates by iteratively applying ABR to the subtrees of a global ABR tree. An **if** condition is incorporated to ensure that each update genuinely improves the global solution: a subtree is replaced only when the newly computed subtree achieves a lower loss than the corresponding region of the current global tree. During each subtree optimization, the remainder of the global tree is held fixed, meaning that the sample assignments to all other leaves remain unchanged. Consequently, whenever a subtree is updated, the predictive performance of its associated leaves is guaranteed to improve or remain unchanged, thereby ensuring that the accuracy of the overall tree monotonically increases.
>
> ## Better positioning w.r.t DPDT
> We agree that the accuracy of ABR and DPDT(Np,) are the same theoretically, however, the efficiency is totally different especially for continuous features. We have added the relative results (**the performance of DPDT(Np) on 51 small datasets**) in Table 1 of revised paper and one experiment as required in Question 2.
>
> ## Reply to Questions:
>  - This result reflects the best test accuracy. Our goal here is to illustrate the potentially best improvement (the best) that can be achieved through careful tuning of $\alpha$. We acknowledge that, in practice, hyperparameter selection should be performed using validation data, and in the final version we may remove this section for that reason. However, cross-validation is computationally expensive, and we cannot provide full results within the rebuttal period. We apologize for this limitation.
> -  Yes, we use the *Avila* dataset as an illustrative example. We have also added the new experiment to analyze the performance and show the results in **figure 7 of the revised paper**. This result illustrates the trade-offs between accuracy, depth, and training time, and we connect points of equal depth to show how depth relates to runtime. Overall, MHABR delivers the strongest accuracy, reaching 99.66% training accuracy at depth $d=10$ and the highest testing accuracy of 98.68%, though its testing accuracy declines at larger depths due to overfitting. In comparison, DPDT and DPDT(Np) only approach this accuracy at $d≥14$, and CART at $d≥17$. While CART is computationally fastest, it requires much deeper trees to be competitive. DPDT improves accuracy with a modest increase in time, but mainly in its default setting. For higher accuracy, MHABR performs better and is faster than DPDT(Np), offering the important advantage of achieving high accuracy with smaller, more interpretable trees. At the same depth, MHABR consistently outperforms DPDT(Np), reflecting the benefit of the MH component on accuracy; it also trains faster before overfitting due to the Reduction technique. Although its runtime grows more quickly beyond depth 13, caused by MH (consistent with our complexity analysis), the model already reaches 100% training accuracy at that point, so additional MH iterations can be relaxed or omitted.
> -  We can change the terminating tolerance $\varepsilon$, which will let the algorithm terminates earlier than evaluating all the candidate splits. It is like the global MIP method, as when the algorithm terminates when the node is small enough, the optimality gap can be estimated (even maybe loose) by the lower bound.

---

> ### Comment · Reviewer_nJuV · 2025-11-25
> **Hyperparameters and cross validations**
>
> I still believe it would strenghten the paper to have a larger scale experiments comparing test or validation accuray on multiple runs on multiple datasets **WITH HYPERPARAMETER search** for trees of depth at most 5.
> Running CART and DPDT with various hyperparameters should be quite fast to do a better experiment:
> Here is the list of hyperparameters from *Why do tree-based models still outperform deep learning on tabular data?, 2022* ,
> *Breiman meets Bellman: Non-Greedy Decision Trees with MDPs, 2025*, *Hyperopt-Sklearn:
> Automatic Hyperparameter Configuration for Scikit-Learn, 2014*:
>
>
> **CART**
> ```python
> config_random = {
>     "model_type": {
>         "value": "sklearn-tree"
>     },
>     # Parameter space taken from Hyperopt-sklearn except when mentioned
>     "model__max_depth": {
>         "values": [5, 2, 3, 4],
>         "probabilities": [0.7, 0.1, 0.1, 0.1]
>     },
>
>     "model__random_state": {
>         "values": [0, 1, 2, 3, 4],
>         "probabilities": [1/5] * 5
>     },
>
>     "model__min_samples_split": {
>         "values": [2, 3],
>         "probabilities": [0.95, 0.05]
>     },
>
>     "model__min_impurity_decrease": {
>         "values": [0.0, 0.01, 0.02, 0.05],
>         "probabilities": [0.85, 0.05, 0.05, 0.05],
>     },
>
>     "model__max_leaf_nodes": {
>         "values": [2**5, 5, 10, 15],
>         "probabilities": [0.85, 0.05, 0.05, 0.05]
>     },
>
>     "model__min_samples_leaf": {  # Not exactly like Hyperopt
>         "distribution": "q_log_uniform_values",
>         "min": 1.5,
>         "max": 50.5,
>         "q": 1
>     },
>
>     "model__min_weight_fraction_leaf": {
>             "values": [0.0, 0.01],
>             "probabilities": [0.95, 0.05]
>         },
>
>     "model__max_features": {
>         "values": ["sqrt", "log2", 10_000],
>         "probabilities": [0.5, 0.25, 0.25]
>     },
> ```
>
> **DPDT**
> ```python
> config_random = {
>     "model_type": {
>         "value": "sklearn-tree"
>     },
>     # Parameter space taken from Hyperopt-sklearn except when mentioned
>     "model__max_depth": {
>         "values": [5, 2, 3, 4],
>         "probabilities": [0.7, 0.1, 0.1, 0.1]
>     },
>
>     "model__random_state": {
>         "values": [0, 1, 2, 3, 4],
>         "probabilities": [1/5] * 5
>     },
>
>     "model__min_samples_split": {
>         "values": [2, 3],
>         "probabilities": [0.95, 0.05]
>     },
>
>     "model__min_impurity_decrease": {
>         "values": [0.0, 0.01, 0.02, 0.05],
>         "probabilities": [0.85, 0.05, 0.05, 0.05],
>     },
>
>     "model__cart_nodes_list": {
>         "values": [(32,), (32, 16), (8,), (8,4,), (4,8,), (16,2,), (4,4,4,), (8,2,2,)],
>         "probabilities": [0.125]*8
>     },
>
>     "model__min_samples_leaf": {  # Not exactly like Hyperopt
>         "distribution": "q_log_uniform_values",
>         "min": 1.5,
>         "max": 50.5,
>         "q": 1
>     },
>
>     "model__min_weight_fraction_leaf": {
>             "values": [0.0, 0.01],
>             "probabilities": [0.95, 0.05]
>         },
>
>     "model__max_features": {
>         "values": ["sqrt", "log2", 10_000],
>         "probabilities": [0.5, 0.25, 0.25]
>     },
> ```

---

> > ### Comment · Reviewer_nJuV · 2025-11-25
> > **Good new experiments that should go in the main paper.**
> >
> > The new experiment on Avila is very good. It should probably go in the main paper after some corrections:
> > - Time in seconds might not be the best indicator of speed per say isnt' MHABR in Julia while DPDT is in Python? What about parallelization, did you deactivate it for both? Isnt there some common operations in CART MHABR and DPDT that you can count?
> > - The speed difference is also most likely due to DPDT(Np,) calling CART with no node limit to getall the roots rather than doing a simple enumeration. But fair enough.
> > - Could you redo this experiment with more repetitions (to get error bars) and at least 1 or 2 more datasets?
> >
> > I have increased my score because this a very good experiment.

---

> ### Comment · Reviewer_nJuV · 2025-11-25
> **Better clarity and schematics**
>
> Thanks. If the better is accepted I trust you will incorporate discussions with reviewers to the best of your abililty. Good job.

---

> ### Author Response · Authors · 2025-11-27
>
> Thank you again for your acknowledgment of our work. Your feedback has been truly invaluable in improving the quality of our experiments. This result will be included in the main body in the final version.
>
> We have added the results of $\alpha$ tuning with validation (50\% for training, 25\% for validation, and 25\%) on 51 small datasets (detailed in Table 9 and Table 10 in appendix A.5)
>
> ----
>
> Yes, we agree that implementation details may affect the running time. We also confirm that **none of the compared methods use parallelization**. Our method does not support parallelization either, as the Reduction procedure is not compatible with it.
>
> Regarding the comparison of common operations, we would like to clarify the following:
> In principle, the computational cost of these methods can be measured by counting the number of depth-1 CART evaluations, since this serves as the basic unit of computation of these methods (for example, a depth-2 CART corresponds to three depth-1 CART calls). However, this comparison is not entirely fair. We uitilized a reduced CART implementation (also a result of this paper, described in Appendix C.2), which is faster than the standard version. In addition, the objective functions differ across implementations, and, scikit-learn does not support "misclassification loss" (used in our method), which further complicates direct equivalence in operation counts.
>
> ----
>
> Furthermore, we have added experiments on the Eeg dataset, including the standard deviation. Actually, selecting datasets for these experiments is not straightforward, as there are specific requirements:
> - The dataset must be sufficiently large and contain continuous features (align with the assumption of our method).
> - The training accuracy should not approach 100% for $d \leq 8$, so that we can observe meaningful trends while avoiding overfitting.
> Given these constraints, we were able to conduct experiments on only these two datasets selected from our processed backup within the limited time available.

---

### Official Review · Reviewer_iD6M · 2025-11-02

**Soundness:** 3
**Presentation:** 3
**Contribution:** 2
**Rating:** 6
**Confidence:** 3

**Summary:**

This paper proposes a bi-level training framework and a branch-and-reduce method for constructing the optimal deep classification trees. The proposed method applies not only to binary features but also to continuous features. To enhance its efficiency for large datasets, this paper introduces an approximate solution for the sub-level problem within the branch-and-reduce framework. Compared with heuristic baselines, the proposed method achieves better accuracy; while compared with global optimal methods, the proposed method maintains scalability on large datasets.

**Strengths:**

1. This paper is well written with clear method description and mathematical notation.

2. This paper prove that the branch-and-reduce method converges to the global optimum of deep classification trees. This paper also introduces an approximate solution for the sub-level problem within the branch-and-reduce framework, which make it practical for large datasets. Thus, this paper strikes a balance between theoretical rigor and practical utility.

3. The experiments are good and the source code is publicly available.

**Weaknesses:**

1. The branch-and-reduce method converges to the global optimum. However, the authors don't provide the convergent speed or computational complexity of this method. Is the computational complexity the same as the brute-force search?

2. Due to the branch nature of decision trees, the idea of branch and reducing (for building decision trees) is not difficult to think of. I’m afraid similar work may already exist, e.g. [1].

[1] Branches: A Fast Dynamic Programming and Branch & Bound Algorithm for Optimal Decision Trees.
Arxiv 2406.02175

**Questions:**

Can this method extend to regression problems or more general objectives?

---

> ### Author Response · Authors · 2025-11-24
>
> We sincerely appreciate your recognition of the presentation quality and technical soundness of our paper. Your concerns are addressed as follows:
>
> ## Reply to Weakness
> 1. The worst case would be the same as brute-force search, however this situation is very rare. The effectiveness by reduction can be observed in the ablation study in Figure 8, Appendix A.6.2. The running time of each ABR is reduced around 100 times. Besides, we add an addition experiments that can further prove this improvement. The tuned DPDT (denoted by DPDT(Np)) can theoretically obtain the same result of ABR without reduction. The Pareto front (Figure 7 in revised paper) of accuracy vs. time show that MHABR has better accuracy with less time, where the accuracy is improved by MH and the efficiency is improved by reduction. Generally, BnR does not have a convergent rate, which is the same as typical branch and bound methods. It is a **tree search**, not an iterative descent, so it does not produce a monotone sequence of approximations. The progress is instance-dependent and highly irregular.
>
> 2. Actually, our method is conceptually developed from QuantBnB. However, [1] targets multi-split sparse trees with binary features (though it can support non-binary features), which constitutes a fundamentally different model structure and optimization setting from ours. In contrast, our approach focuses on deep, binary-split optimal trees with continuous features, leading to an entirely different optimization problem. Moreover, the reduction mechanism in our BnR component is specifically designed for continuous features and plays a crucial role in improving computational efficiency, as demonstrated in the ablation study in Appendix A.6.2. The Pareto front (Figure 7 in revised paper) of accuracy vs. depth show that DPDT(Np) does not provide too much improvement, while MHABR can obtain much better accuracy at the same depth.
>
> ## Reply to Questions:
> - Yes, this method can be extended to regression problems as well as to boosting. To  regression problems, one can change the objective value to MSE. In terms of boosting, it maybe time-consuming compared to CART and DPDT.

---

> > ### Author Response · Authors · 2025-11-28
> >
> > Dear Reviewer,
> >
> > We have revised the paper since the last update in response to the comments from Reviewers FhND and nJuV.
> >
> > The updates are reflected in the Introduction (paragraphs 2 and 3) and in Appendix A.5 (Figure 7, Figure 8, Table 9, and Table 10).
> >
> > Please refer to the latest version of the paper if you would like to provide additional comments.
> >
> > We sincerely appreciate the time and effort you have dedicated to our work.
> >
> > Best regards,
> >
> > The authors

---

### Official Review · Reviewer_gxZq · 2025-11-08

**Soundness:** 3
**Presentation:** 3
**Contribution:** 3
**Rating:** 4
**Confidence:** 3

**Summary:**

This paper developed a hybrid approach to learning decision trees that attempts to bridge the gap between fast, greedy heuristics (like CART) and computationally intractable optimal methods (like MIP or full dynamic programming).

**Strengths:**

1. The core idea is to use a branch-and-reduce framework for the top of the tree while approximating the value of deeper subproblems with a fast heuristic. I think this is clever and draws from well-established principles in optimization and control theory (approximate dynamic programming).

2. The amount of experiments is sufficient and I think the authors have demonstrated some improvements over numerous baselines in terms of accuracy and scalability.

**Weaknesses:**

1. The entire framework is built on the premise that you can approximate the "optimal tail problem" (the lower-level problem, LLP) using CART (Line 240). Yet, CART is the quintessential myopic, greedy heuristic that this entire line of research seeks to improve upon. How can a method that uses a demonstrably poor, one-step-optimal heuristic as its core value function for future states ever hope to achieve near-optimality for deep trees? Aren't you simply performing a more computationally expensive, one-step lookahead from the root, built upon the very same greedy foundation you criticize?

2. Your Section 3.2 attempts to frame the approach from a RL perspective, referencing DPDT and describing your LLP approximation as a "lookahead rollout". I think this framing is superficial and it literatually adds little substance to the method. A rollout is a Monte Carlo simulation from a state, whereas you are simply replacing a value function $V^\ast$ with a heuristic $V_\text{heuristic}$. Is this not just standard approximate dynamic programming? What specific, novel insights or algorithmic components does the RL framing provide that are not already well-understood concepts from DP and heuristic search?

3. One thing I think is not a good practice is that the authors repeatedly emphasizes the proposed ABR method "guarantees global optimality for depth-2 trees" (Lines 260-262) and highlights this in the contributions (Line 084). Given that CART itself is optimal for depth-1 trees, isn't it a trivial and expected outcome that a method performing a full branch-and-bound search at the root (ULP) and using an optimal depth-1 solver for the leaves (LLP) would be optimal for depth-2? How is this anything more than a basic sanity check of your framework, and why is it presented as a significant contribution?

4. Your key efficiency gain comes from the "branch-and-reduce" method, which prunes the search space of split points. In the full BR method, this is sound because the lower bound is valid. However, in your approximate ABR method, you acknowledge that the continuity condition may not hold and the reduction strategy can be "inaccurate" (Lines 330-332), and in the limitations, you concede that "unpredictable reduction errors...make formal guarantees...difficult to achieve" (Lines 472-474). If the reduction step can erroneously prune branches that contain the true optimal split, what prevents your algorithm from catastrophic failure? I think this may lead to a solution that is arbitrarily bad, potentially even worse than CART?

5. From my understanding, the proposed MHABR algorithm is a highly complex! It has multi-stage process involving a bilevel formulation, an outer loop of Moving Horizon iterations, and an inner loop of ABR, which itself contains branching, bounding, and reduction steps. While you demonstrate scalability improvements over exact methods, have you not simply created an enormously complicated and over-engineered heuristic that achieves only marginal gains over much simpler, state-of-the-art heuristics like DPDT or tuned versions of CART, especially when considering the significant implementation and computation overhead?

**Questions:**

See my comments in the Weaknesses part.

---

> ### Author Response · Authors · 2025-11-24
>
> Thank you for your efforts and targeted comments. We share your concerns about the possibility of over-engineered heuristics and fully understand the issues you raised. In fact, the algorithm itself is not complex, its description may appear detailed, but the underlying procedure is straightforward. We present it rigorously for completeness. Our responses to these concerns are provided below:
>
> the your questions and concerns are replied as follow:
> 1. While it is true that CART guarantees optimality only for depth-1 trees, this does not imply that it cannot be leveraged to guide the search for deeper optimal trees. DPDT serves as a prominent example: it builds upon CART and can be configured to approach the globally optimal tree by appropriately improving the action (candidate split) space. Although our method is computationally more intensive than CART, it achieves substantially higher accuracy.
> 	For many optimization methods, the core idea is to solve a complex problem through a sequence of simpler subproblems. For example, in sequential quadratic programming (SQP), each iteration solves a quadratic program to obtain a descent direction, and the solution is updated accordingly.
> Besides, we have compared to CART, DPDT, and DPDT(Np)(a new version of DPDT with (cart_nodes_list=(Np, )). The results are shown in Figure 7 (Appendix A.4) in revised paper.
> In the Pareto front of accuracy vs. tree depth, we can see that MHABR attains the same training accuracy (100%) with significantly smaller depth, resulting in a much simpler model compared to both CART and DPDT.
>
> 2. In Section 3.2, we employ concepts from ADP to illustrate the bilevel structure of our method, aiming to help reviewers better understand the approach, similar to how DPDT presents its framework. Our method is not intended to advance ADP or develop new ADP techniques; rather, we show an RL perspective on our approach, which also provides an intuitive explain about our algorithm. This perspective allows others to interpret and potentially extend our method using reinforcement learning principles in future work. You raised concerns about the method appearing over-engineered, this subsection is exactly intended to clarify the underlying idea in a simple and accessible way.
>
> 3. This demonstrates that our algorithm can be extended to global optimal methods while maintaining relatively high efficiency (see Fig.~3). Optimality is a central concern in evaluating optimization methods. Our results show that in certain cases (e.g., $d=2$ for ABR, $d=3$ for ABR$^2$), our method reaches global optimal solutions, whereas many other near-optimal methods cannot guarantee this. All the near-optimal methods generally aim to reduce the gap to the true optimum, our method effectively closes this gap in these special cases. Furthermore, for any subtrees of depth $3$ obtained by ABR (for any global tree depth $d \geq 3$), MH computes an optimal depth-$2$ tree, which is guaranteed to perform at least as well as, and typically better than, CART.
>
> 4. This issue can be addressed by applying a CART warm start before running ABR, similar to TAO, which guarantees a solution at least as good as CART, as discussed in Section 3.4. We tested this option (not included in the paper), but it did not yield any meaningful improvement in practice. In fact, the performance gain primarily comes from the MH structure, as demonstrated in our ablation study (Appendix A.6.2). We also added an additional experiment using a tuned DPDT configuration (cart_nodes_list = (Np,)), which can theoretically get the same solution as ABR without Reduction. The results further confirm this conclusion: although tuned DPDT(Np) is theoretically capable of matching ABR (without Reduction), it does not produce substantial accuracy gains in practice, whereas incorporating MH provides a clear and significant improvement, as shown in Figure 7 of the revised paper.
>
> 5. As illustrated by Reviewer nJuV, ABR is somewhat similar to DPDT with specific configuration (cart_node=(Np,)). In this case, our method only adds reduction and MH, where reduction utilizes Theorem 3.1 to reduce the searching space with pretty simple cost ($\mathcal{O}(1)$), and MH improves a lot on accuracy (see ablation study A.6.2, figure 8. (a), from 74.28\% to 93.42\%) with cost of $\frac{d(d-1)}{2}$ repeating ABR of smaller trees.  Even though, CART and DPDT (light version) can be extremely fast, it sacrifices the accuracy. This why many methods like QuantBnB, ConTree, and SPLIT, etc., are proposed.

---

> ### Author Response · Authors · 2025-11-28
>
> Dear Reviewer,
>
> We have revised the paper since the last update in response to the comments from Reviewers FhND and nJuV.
>
> The updates are reflected in the Introduction (paragraphs 2 and 3) and in Appendix A.5 (Figure 7, Figure 8, Table 9, and Table 10).
>
> Please refer to the latest version of the paper if you would like to provide additional comments.
>
> We sincerely appreciate the time and effort you have dedicated to our work.
>
> Best regards,
>
> The authors

---

### Official Review · Reviewer_FhND · 2025-11-08

**Soundness:** 2
**Presentation:** 2
**Contribution:** 3
**Rating:** 4
**Confidence:** 4

**Summary:**

The paper presents Moving-Horizon Approximate Branch-and-Reduce, a novel approach for near-optimal decision trees. It considers the learning of an optimal decision tree as a bi-level optimization problem and proposes a branch-and-reduce technique to solve the problem, and in particular an approximation of the low-level problem using CART, akin to a lookahead rollout in reinforcement learning. Experiments on 51 small datasets as well as 6 medium to large datasets show the proposed approach outperforms heuristic baselines and exhibits better scalability compared to globally optimal approaches.

**Strengths:**

Strengths:
- Novel approach for near-optimal decision trees based on bi-level formulation with a branch-and-reduce solving technique and CART-based approximation of the low-level problem
- The view of the low-level problem approximation as an lookahead rollout in reinforcement learning provides an interesting perspective on the approach
- The numerical experiments show the proposed approach outperforms the heuristic approaches while remaining more scalable than globally optimal approaches.

**Weaknesses:**

Weaknesses:
- Framing of previous literature in the paper: the paper makes various claims about previous work that I do not believe to be accurate
	1. The paper claims that DP- and SAT-based approaches face inherent drawback due to the need to binarize features. However, recent DP-based and SAT-based approaches that do not require binarization and handle continuous features directly have been proposed and not cited: [1] for DP-based, and [2] for SAT-based.
	2. "TAO remains restricted to optimizing threshold values of internal nodes while keeping feature selections fixed”: I don’t believe this is true. A full node (variable and threshold) is fit at every step. From the original paper: "For axis-aligned trees, it can be solved exactly by enumeration over features and splits, just as in the CART algorithm to minimize the impurity" [3].
	3. DPDT’s "reliance on heuristic optimization severely limits its optimality”: I believe DPDT is using dynamic programming for optimization. It does use heuristic approach, specifically CART, to reduce the space of splits it considers (but, as described in the paper, it can be used with an exhaustive function which will maintain optimality). That seems very similar to the proposed approach’s use of CART, and similar to the proposed approach, DPDT is demonstrating “near-optimal” performance.
	4. I think Table 1 is somewhat misleading in terms of its designation of methods as “globally optimal in special cases”. I don't think being optimal only on binary data is similar to being optimal up to depth 2 (they are currently marked similarly in the table). Every dataset can be converted to a binary data without loss of optimality (but with a potentially very large number of features).
- Missing highly-relevant works: there are various “near-optimal” recent approaches that are similar to the proposed approach in the sense of they are relaxation of optimal approaches. They should be cited and considered as baselines in the paper. Some important examples: [4] [5] [6].
- Experimental evaluation:
	1. Not clear why hyper parameters were not tuned for other approaches like DPDT. In fact, I am concerned that the comparison is not really fair: we can see that DPDT finishes order of magnitude faster than MHABR, however DPDT can trade-off more run time for better optimality by considering more splits, meaning a much more extensive configuration could have been considered for DPDT to improve its performance while maintaining lower runtime compared to the proposed approach.
	2. Not clear why other non-greedy / near-optimal approaches are not considered like TAO, SPLIT, top-B, etc. (see list above). These are particular useful as they are the most relevant approaches to this paper and, similar to the proposed approach, tend to outperform heuristic baselines while remaining more scalable than globally optimal approaches.
	3. As the paper notes, much of the improvement in performance comes from the tuning of the regularization. Importantly, other optimal and near-optimal approaches allow for tuning a regularization parameter like GOSDT [7] and SPLIT [4], TAO [3], Branches [8], etc.

Minor:
- Eq. (1) - should be argmin? We would like to find the tree that minimizes the cost function
- How is the complexity of trees (C) measured? Number of nodes?

[1] Brița, C. E., van der Linden, J. G., & Demirović, E. (2025, April). Optimal Classification Trees for Continuous Feature Data Using Dynamic Programming with Branch-and-Bound. In Proceedings of the AAAI Conference on Artificial Intelligence (Vol. 39, No. 11, pp. 11131-11139).

[2] Shati, P., Cohen, E., & McIlraith, S. A. (2023). SAT-based optimal classification trees for non-binary data. Constraints, 28(2), 166-202.

[3] Carreira-Perpinán, M. A., & Tavallali, P. (2018). Alternating optimization of decision trees, with application to learning sparse oblique trees. Advances in neural information processing systems, 31.

[4] Babbar, V., McTavish, H., Rudin, C., & Seltzer, M. (2025). Near Optimal Decision Trees in a SPLIT Second. arXiv preprint arXiv:2502.15988.

[5] Blanc, G., Lange, J., Pabbaraju, C., Sullivan, C., Tan, L. Y., & Tiwari, M. (2023). Harnessing the power of choices in decision tree learning. Advances in Neural Information Processing Systems, 36, 80220-80232.

[6] Kiossou, H. S., Nijssen, S., & Schaus, P. (2025). A Generic Complete Anytime Beam Search for Optimal Decision Tree. arXiv preprint arXiv:2508.06064.

[7] Lin, J., Zhong, C., Hu, D., Rudin, C., & Seltzer, M. (2020, November). Generalized and scalable optimal sparse decision trees. In International conference on machine learning (pp. 6150-6160). PMLR.

[8] Chaouki, A., Read, J., & Bifet, A. (2024). Branches: Efficiently Seeking Optimal Sparse Decision Trees with AO. arXiv preprint arXiv:2406.02175.

**Questions:**

See my questions and concerns under weaknesses above

---

> ### Author Response · Authors · 2025-11-24
> **Theoretical Reply to Weakness (1)**
>
> We appreciate your thoughtful review and constructive comments. We fully understand your concerns regarding the fairness of our experimental comparisons, which we also take very seriously. Following your suggestions, we have revised the relevant sections accordingly. Our detailed responses to your concerns are provided below.
>
> ## Reply to weakness:
> ### **Framing of previous literature**
> 1. Yes, we lost the recently published related work [1], but on the other hand, we believe that [2] is not suitable for inclusion. We clarify below why these omissions do not affect our claims. Work [1] is essentially an extension of QuantBnB. Although the authors present it within a dynamic programming framework, the method inherits the similar limitations (depth $\leq$ 4) as QuantBnB (a MIP-based approach). In particular, its performance on deeper trees does not outperform other binarization-based DP methods, such as STreeD. Regarding [2], it is a SAT-based encoding method rather than a genuine decision-tree learning algorithm. Therefore, [2] is not an appropriate baseline for our setting.
>
> 2. We agree that the statement in [3] is correct. However, even greedy methods such as CART can be extended to improve accuracy by incorporating look-ahead to deeper levels. In our discussion, we focus on the default version of these methods (e.g., CART) rather than these enhanced variants, as the results of enhanced version are not reported and the default implementation is the most representative of what the authors recommend for practical use. Moreover, the authors of DPDT adopts the same viewpoint towards TAO (see Section 2 of DPDT:"This is different from the line of work on Tree Alternating Optimization (TAO) algorithm [12, 13, 61] that only optimizes tree nodes threshold values for fixed nodes features similarly to optimizing neural network weights with gradient-based methods.").
>
> 3. Yes, all of these methods could, in principle, be extended to achieve global optimality by incorporating brute-force search. However, in practice this is never done because the computational cost would be prohibitively high. In contrast, our contribution is to improve efficiency rather than to “solely” pursue optimality through exhaustive search.
>
> 4. We agree with your comment, we initially aimed to simplify the table as much as possible. However, following Reviewers' feedback, we have decided to remove the table entirely. Although our intention was to provide a concise summary of the capabilities of the compared algorithms, the table appears to introduce unnecessary confusion.
>
>
>
> ### **Missing highly-relevant works**:
> 1. Thanks for your comments. We have added the relevant references in the updated version of the paper. Regarding the selection of baselines mention in the whole comments, we would like to clarify our rationale:
> **Clarify the selection of baselines:**
> - **TAO** does not provide code for reproducibility, and has been evaluated by QuantBnB, showing limited improvement. We could only generate trees with the same configuration as reported. The results that show superiority of MHABR are show later.
> - **Top-k [5]** has already been evaluated in the DPDT study and somehow defeated by DPDT (DPDT also refers to Top-k for most informative splits); thus, the comparison to DPDT is better.
> - **SPLIT** and **GOSDT**: These two methods target sparse decision trees designed for binary features and binary labels. Since SPLIT is an extension of GOSDT and has been shown to deliver stronger performance, our comparisons focus primarily on SPLIT. Although both methods can be adapted to continuous features through binarization (in the same manner as DL8.5), the resulting tree structures tend to be sparse and unbalanced, which differs substantially from the balanced trees produced by our approach. Consequently, it is not meaningful to assert that one method universally dominates the other, as their performance may vary depending on the setting. A detailed example is provided later.
> - **Branches** is a method for multi-split tree, while all the other methods mentioned here are binary split methods. This method targets multi-split sparse trees with binary features (though it can support non-binary features), which constitutes a fundamentally different model structure and optimization setting from ours.
> - **The approach in [6]** was posted very recently (August 8, 2025) as an informal arXiv preprint. For fairness and scientific rigor, we believe it is more appropriate to compare against it after the work has undergone peer review and formal acceptance.

---

> ### Author Response · Authors · 2025-11-24
> **Reply to Experimental Evaluation (2)**
>
> ### **Experimental evaluation:**
> 1. We agree with your suggestion, we have add the experiments that compares tuned DPDT to compare the testing results and revised the table. Besides, we add a new version of DPDT with configuration of cart_nodes_list=(Np, ), using the dataset *Avila* to plot the Pareto front (accuracy vs. depth and accuracy vs. train time). The results are shown in Appendix.A.4 （Figure.7) of revised version.
>
> 2. We have noted these methods, however, they are not very appropriate to be compared to as we explained above (reply to Missing highly-relevant works). Therefore, we add the experiments on SPLIT and TAO:
> - **Compared to SPLIT**
> We use the **Avila** dataset to illustrate how the parameter choices affect the accuracy of both methods. For SPLIT, we compare performance across different values of $\alpha \in \{0.001, 0.005, 0.01\}$, as SPLIT achieves better accuracy when $\alpha \leq 0.01$, and across maximum depths from 3 to 8. Although SPLIT includes a parameter for _lookahead depth_, its impact on accuracy is reported to be limited in its original paper, so we do not adjust it here. For fairness, since SPLIT produces at most a tree with approximately 30 splits on this dataset, we restrict MHABR to depths $d \leq 5$, for which it produces 31 splits. The comparison results are shown below:
> #### SPLIT
>
> | alpha | max_depth | lookahead_depth | train accuracy | test accuracy | splits | time  |
> | ----- | --------- | ------- | -------------- | ------------- | ------ | ----- |
> | 0.001 | d=3       | 2  | 49.20          | 48.99         | 2.20   | 4.38  |
> | 0.001 | d=4       | 2      | 53.18   | 53.13         | 6.00   | 4.34  |
> | 0.001 | d=5       | 2    | 55.73          | 55.35         | 12.60  | 4.47  |
> | 0.001 | d=6       | 2    | 58.11          | 57.11         | 22.30  | 5.26  |
> | 0.001 | d=7       | 2               | 59.55          | 57.98         | 29.20  | 8.49  |
> | 0.001 | d=8       | 2               | 59.82          | 58.32         | 30.30  | 19.02 |
> | 0.005 | d=3       | 2               | 49.16          | 49.02         | 2.00   | 4.35  |
> | 0.005 | d=4       | 2               | 53.18          | 53.13         | 6.00   | 4.51  |
> | 0.005 | d=5       | 2               | 54.06          | 53.80         | 7.30   | 4.57  |
> | 0.005 | d=6       | 2               | 54.75          | 54.34         | 8.30   | 5.02  |
> | 0.005 | d=7       | 2               | 54.81          | 54.37         | 8.40   | 6.98  |
> | 0.005 | d=8       | 2               | 54.81          | 54.37         | 8.40   | 10.90 |
> | 0.010 | d=3       | 2               | 49.16          | 49.02         | 2.00   | 4.28  |
> | 0.010 | d=4       | 2               | 51.08          | 50.95         | 3.00   | 4.34  |
> | 0.010 | d=5       | 2               | 51.08          | 50.95         | 3.00   | 4.34  |
> | 0.010 | d=6       | 2               | 51.08          | 50.95         | 3.00   | 4.72  |
> | 0.010 | d=7       | 2               | 51.08          | 50.95         | 3.00   | 5.76  |
> | 0.010 | d=8       | 2               | 51.08          | 50.95         | 3.00   | 7.39  |
>
> ## MHABR
> | α      | depth | train acc | test acc | splits | time |
> |--------|-------|-----------|----------|--------|------|
> | 0.001  | 2     | 54.08     | 54.38    | 3      | 0.58 |
> | 0.001  | 3     | 58.50     | 58.42    | 7      | 2.14 |
> | 0.001  | 4     | 61.59     | 60.75    | 15     | 5.30 |
> | 0.001  | 5     | 67.76     | 66.34    | 31     | 7.13 |
> | 0.005  | 2     | 54.08     | 54.38    | 3      | 0.58 |
> | 0.005  | 3     | 58.50     | 58.42    | 7      | 2.15 |
> | 0.005  | 4     | 61.59     | 60.75    | 15     | 5.23 |
> | 0.005  | 5     | 67.76     | 66.34    | 31     | 7.07 |
> | 0.01   | 2     | 54.08     | 54.38    | 7      | 0.56 |
> | 0.01   | 3     | 58.49     | 58.47    | 7      | 2.13 |
> | 0.01   | 4     | 61.61     | 60.80    | 15     | 5.32 |
> | 0.01   | 5     | 67.76     | 66.34    | 31     | 7.13 |
>
> We observe that when $d \geq 6$, the accuracy of SPLIT does not increase noticeably for any value of $\alpha$. Its best testing accuracy is $58.32\%$ with 30.3 splits, whereas MHABR can achieve the same accuracy using only 7 splits. Moreover, MHABR attains $66.34\%$ testing accuracy at depth 5, which is substantially higher than that of SPLIT. However, an examination of the resulting tree structures shows that SPLIT tends to produce unbalanced trees, while MHABR is more likely to generate balanced trees. Therefore, although MHABR achieves higher accuracy, we can only claim that our method exhibits better optimality than SPLIT for deep trees (specifically $d\geq 6$).

---

> ### Author Response · Authors · 2025-11-24
> **Reply to Experimental Evaluation and Minor (3)**
>
> ### Compared to TAO:
> As TAO does not provide code for reproducibility, we evaluated our method on the same five datasets reported in Appendix 2.1 of TAO, using the same data split (50% training, 25% validation, 25% testing). The corresponding results are presented below:
>
> | Dataset (depth)                  | TAO train | TAO test | MHABR train | MHABR test |
> |----------------------------------|-----------|----------|-------------|------------|
> | Balance-scale (d=2)                | 72.5 (1.6) | 69.5 (2.9) | 72.7 (0.7)  | 68.5 (2.0) |
> | Balance-scale (d=3)                | 76.9 (1.1) | 71.6 (1.9) | 76.8 (1.4)  | 74.8 (2.2) |
> | Balance-scale (d=4)                | 84.0 (1.5) | 79.8 (3.1) | 84.1 (1.1)  | 78.9 (2.7) |
> | Banknote-authentication (d=2)      | 91.9 (0.4) | 90.6 (0.9) | 92.8 (0.4)  | 91.3 (1.1) |
> | Banknote-authentication (d=3)      | 96.0 (0.5) | 95.7 (1.2) | 98.3 (0.4)  | 97.2 (0.6) |
> | Banknote-authentication (d=4)      | 98.9 (0.7) | 97.2 (0.7) | 99.5 (0.3)  | 98.7 (0.5) |
> | Blood-transfusion (d=2)            | 78.0 (0.8) | 75.8 (2.0) | 77.9 (1.4)  | 77.1 (3.7) |
> | Blood-transfusion (d=3)            | 79.5 (1.0) | 77.0 (2.0) | 80.3 (1.4)  | 77.8 (2.9) |
> | Blood-transfusion (d=4)            | 81.6 (1.3) | 77.2 (1.3) | 80.7 (2.3)  | 77.6 (3.5) |
> | Breast-cancer-diagnostic (d=2)     | 95.0 (0.5) | 92.7 (2.2) | 96.4 (0.4)  | 93.9 (1.7) |
> | Breast-cancer-diagnostic (d=3)     | 97.0 (0.6) | 93.1 (1.4) | 98.5 (0.5)  | 95.0 (1.7) |
> | Breast-cancer-diagnostic (d=4)     | 98.0 (0.5) | 93.2 (0.5) | 99.6 (0.3)  | 95.1 (1.8) |
> | Spambase (d=2)                     | 86.5 (0.7) | 86.1 (1.0) | 87.3 (0.5)  | 86.9 (0.6) |
> | Spambase (d=3)                     | 90.0 (0.4) | 89.1 (1.0) | 90.8 (0.3)  | 90.3 (0.8) |
> | Spambase (d=4)                     | 91.8 (0.3) | 90.3 (0.8) | 92.3 (0.6)  | 91.3 (0.8) |
>
> Overall, the average training accuracy of TAO is 87.8%, whereas MHABR achieves 88.5%. For testing, TAO reaches 85.3% while MHABR attains 86.3%. MHABR consistently outperforms TAO, showing at least a 0.7% improvement across the five datasets at depths 2, 3, and 4. Based on this trend, we expect that MHABR will demonstrate even greater gains at depth $d=8$, as the performance gap between the two methods increases with tree depth. Unfortunately, TAO does not report results for this setting, and we are unable to reproduce them fairly without access to its implementation.
>
>
> ## Reply to Minor:
> - No, it should be a minimization (in Eq. (1)). We apologize for the lack of clarity in the original text. The function $V_d$ represents the optimal objective value (loss), corresponding to the solution of the optimization problem $\min L(\mathcal{I}, T)$. In this optimization problem, the dataset $\mathcal{I}$ and the tree $T$ serve as the arguments, where $T$ is the decision variable and $\mathcal{I}$ is fixed for a given dataset. However, $\mathcal{I}$ varies when considering a subtree, as it becomes the subset of samples that reach that subtree.
>
> We have now added an explicit 'argmin' equation to denote the optimization problem used to find the optimal tree. It was not a typographical error: we intended to use $V_d$ to represent the loss (value or misclassification). However, we agree that the notation was unclear, and we have revised this part of the manuscript for improved readability. For line 149, we have added the references in the revised version.
> $$T^*(\mathcal{I}) = \arg\min_{T\in\mathcal{T}_d} L(\mathcal{I},T)$$
> - Yes, the complexity of a decision tree is measured by the number of branch nodes.

---

> > ### Comment · Reviewer_FhND · 2025-11-27
> >
> > Thank you for the additional results. I think they would indeed strengthen the paper. And thanks for providing clarifications.
> >
> > Regarding my points under "Framing of previous literature", my concerns are about what I think is inaccurate/incomplete framing of existing literature in the paper:
> > - For example, the statement that DP and SAT based approaches require binarization (which is not accurate as recent works in both areas have proposed formulations that are not based on binarizaiton).
> > - TAO is not restricted to optimizing thresholds and can be used to optimize the features as well in axis-aligned trees.
> > - DPDT's reliance on heuristic optimization to speed up performance is not fundamentally different than other approaches (including the one proposed in the paper) that utilize heuristic optimization as a component that can improve performance, and it is indeed able to obtain near optimal solutions.
> >
> > These are not criticisms of the proposed approach or requests to include additional baseline. These are concerns about the framing of existing literature. I am therefore not sure I understand the authors' response on this, that does not dispute the correctness of my concerns about the current framing but also does not suggest to change the framing to more accurately reflect the existing literature.

---

> > > ### Author Response · Authors · 2025-11-28
> > >
> > > We sincrely appreciate your reply. We apologize for the earlier inaccurate framing. We have revised the relevant sections in the paper according to your suggestions. The detailed changes are as follows:
> > >
> > > ----
> > >
> > > ###  DP and SAT based approaches:
> > > we have added:
> > >
> > > "It should be noted that although ConTree [1] is based on DP and MIP, it is built on Quant-BnB and, like other MIP-based methods, remains limited to a maximum depth of 4."
> > >
> > > "some methods directly consider encoding DTs [2]"
> > >
> > > and we point out that for "**most** DP and SAT based approaches" they "**often**" need binarization.
> > >
> > > ----
> > >
> > > ### TAO
> > > We have removed the sentences about TAO in axis-aligned trees, including:
> > >
> > > "However, TAO remains restricted to optimizing threshold values of internal nodes while keeping feature selections fixed."
> > >
> > > Instead, we simply note that TAO provides only modest improvements over CART, as supported by prior work and by the experiments we added in our earlier response.
> > >
> > > ----
> > >
> > > ### DPDT
> > > In terms of DPDT, we have deleted:
> > >
> > > "Its reliance on heuristic optimization severely limits its optimality"
> > >
> > > and added:
> > >
> > > "its further accuracy gains come at a substantial computational cost, which significantly limits its optimality. Within practical time budgets, the trees it produces remain far less accurate than globally optimal ones."
> > >
> > > This can be proved by additional experiments of Figure 7 in revised paper.

---

### Meta-Review · Area_Chair_AcHX · 2025-12-16

**Summary:**

The paper proposes MHABR, a bilevel framework for near-optimal deep classification trees, combining branch-and-reduce with a heuristic lower-level solver. Reviewers recognized its technical soundness, strong empirical performance, and relevance for deeper trees. Concerns included inaccurate literature framing, limited / unfair baseline comparisons, and unclear presentation.

**Reviewer Concerns:**

Addressed by the rebuttal and revisions:
- Inaccurate framing of prior work (DPDT, TAO, DP/SAT methods) was acknowledged and corrected, with problematic statements removed or softened.
- Misleading summary tables and figures were removed, and new experiments were added to better illustrate accuracy–runtime–depth trade-offs.
- Fairness of comparisons was improved via additional experiments.
- Clarity issues in notation (e.g., argmin formulation), algorithm description, and MH refinement were improved.
Reviewer requests for stronger positioning relative to DPDT were explicitly addressed in both text and experiments.

Partially or still outstanding:
- Some concerns remain that the added algorithmic complexity is not fully justified relative to simpler near-optimal methods, especially given implementation overhead.
- The evaluation setting remains unclear due to the absence of a standardized protocol, particularly concerning statistical significance and a hyperparameter optimization procedure using predefined grids from related work and existing benchmarks.

**Reviewer Scores:**

Reviewer `[FhND]` responded to the rebuttal but did not indicate that their concerns were fully resolved, so even if the discussion period had not ended early, I would not expect a score increase.
Reviewer `[nJuV]` indicated a score increase (most likely from borderline reject (4) to borderline accept (6)), mainly based on the new experiment on Avila. I do not expect the reviewer to raise the score further, even though the discussion continued, since concerns regarding the HPO were not fully addressed.
The other reviewers did not engage in the rebuttal, and I do think that both had maintained their original ratings of 4 (`[gxZq]`) and 6 (`[iD6M]`).
Therefore, even after the rebuttal, the paper would remain highly borderline,  most likely with a score of 6644, and without a reviewer championing the paper or clearly suggesting acceptance.

---

### Decision · Program_Chairs · 2026-01-26

Reject